# Human VMPFC encodes early signatures of confidence in perceptual decisions

Sabina Gherman, Marios G. Philiastides*

Institute of Neuroscience and Psychology, University of Glasgow, Glasgow, United Kingdom

**Abstract** Choice confidence, an individual's internal estimate of judgment accuracy, plays a critical role in adaptive behaviour, yet its neural representations during decision formation remain underexplored. Here, we recorded simultaneous EEG-fMRI while participants performed a direction discrimination task and rated their confidence on each trial. Using multivariate single-trial discriminant analysis of the EEG, we identified a stimulus-independent component encoding confidence, which appeared prior to subjects' explicit choice and confidence report, and was consistent with a confidence measure predicted by an accumulation-to-bound model of decision-making. Importantly, trial-to-trial variability in this electrophysiologically-derived confidence signal was uniquely associated with fMRI responses in the ventromedial prefrontal cortex (VMPFC), a region not typically associated with confidence for perceptual decisions. Furthermore, activity in the VMPFC was functionally coupled with regions of the frontal cortex linked to perceptual decision-making and metacognition. Our results suggest that the VMPFC holds an early confidence representation arising from decision dynamics, preceding and potentially informing metacognitive evaluation.

DOI: https://doi.org/10.7554/eLife.38293.001

*For correspondence: marios.philiastides@glasgow.ac.uk

Competing interests: The authors declare that no competing interests exist.

## Introduction

Our everyday lives involve situations where we must make judgments based on noisy or incomplete sensory information – for example deciding whether crossing the street on a foggy morning, in poor visibility, is safe. Being able to rely on an internal estimate of whether our perceptual judgments are accurate is fundamental to adaptive behaviour and accordingly, recent years have seen a growing interest in understanding the neural basis of confidence judgments.

Within the perceptual decision making field, several studies have sought to characterise the neural correlates of confidence during metacognitive evaluation (i.e., while subjects actively judge their performance following a choice), revealing the functional involvement of frontal networks, in particular the lateral anterior and anterior cingulate prefrontal cortices (*Fleming et al., 2012*; *Hilgenstock et al., 2014*; *Morales et al., 2018*). Concurrently, psychophysiological work in humans and non-human primates using time-resolved measurements has shown that confidence encoding can also be observed at earlier stages, and as early as the decision process itself (*Kiani and Shadlen, 2009*; *Zizlsperger et al., 2014*; *Gherman and Philiastides, 2015*).

In line with these latter observations, recent fMRI studies have reported confidence-related signals nearer the time of decision (e.g., during perceptual stimulation) in regions such as the striatum (*Hebart et al., 2016*), dorsomedial prefrontal cortex (*Heereman et al., 2015*), cingulate and insular cortices (*Paul et al., 2015*), and other areas of the prefrontal, parietal, and occipital cortices (*Heereman et al., 2015*; *Paul et al., 2015*). Interestingly, confidence-related processing has also been reported in the ventromedial prefrontal cortex (VMPFC) during value-based decisions and various ratings tasks (*De Martino et al., 2013*; *Lebreton et al., 2015*), however the extent to which this

**eLife digest** While waiting to cross the road on a foggy morning, you see a shape in the distance that appears to be an approaching car. How do you decide if it is safe to cross? We often have to make important decisions about the world based on imperfect information. What guides our subsequent actions in these situations is a sense of accuracy, or confidence, that we associate with our initial judgments. You would not step off the kerb if you were only 10% confident the car was a safe distance away. But how, when, and where in the brain does such confidence emerge?

Gherman and Philiastides examined how brain activity relates to confidence during the early stages of decision-making, that is, before people have explicitly committed to a particular choice. Healthy volunteers were asked to judge the direction in which dots were moving across a screen. They then had to rate how confident they were in their decision. Two techniques – EEG and fMRI – tracked their brain activity during the task. EEG uses scalp electrodes to reveal when and how electrical activity is changing inside the brain, while fMRI, a type of brain scan, shows where these changes in brain activity occur. Used together, the two techniques provide a greater understanding of brain activity than either used alone.

Activity in multiple regions of the brain correlated with confidence at different stages of the task. Certain brain networks showed confidence-related activity while the volunteers tried to judge the direction of movement, and others were engaged when volunteers made their confidence ratings. However, activity in only one area reliably indicated how confident the volunteers felt before they had made their choice. This area, the ventromedial prefrontal cortex, also helps process rewards. This suggests that feelings of confidence early in the decision-making process could guide our behaviour by virtue of being rewarding.

Many brain disorders – including depression, schizophrenia and Parkinson's disease – compromise decision-making. Patients show changes in accuracy, response times, and in their ability to accurately evaluate their decisions. The methods used in the current study could help reveal the neural changes that cause these impairments. This could lead to new methods to diagnose and predict cognitive deficits, and new ways to treat them at an earlier stage.
DOI: https://doi.org/10.7554/eLife.38293.002

region is additionally involved in perceptual judgments relying on temporal integration of sensory evidence remains unclear.

Importantly, the studies above suggest that confidence is likely to involve a temporal progression of neural events requiring the involvement of multiple networks, as opposed to a single event or quantity. Identifying neural confidence representations that arise early in the decision process (e.g., prior to metacognitive report or as early as the choice itself) is an important prerequisite in understanding the broader confidence-related dynamics, as these signals may provide the basis for higher-order and more deliberate processes such as metacognitive appraisal. Nevertheless, efforts to characterise early confidence representations in the human brain have been limited.

One potential limitation in previous approaches to studying the neural representations of confidence is the exclusive reliance on correlations with behavioural measures, most commonly in the form of subjective ratings given by participants after the decision (*Grimaldi et al., 2015*). However, theoretical and empirical work suggests that post-decisional metacognitive reports may be affected by processes occurring after termination of the initial decision (*Resulaj et al., 2009*; *Pleskac and Busemeyer, 2010*; *Fleming et al., 2015*; *Moran et al., 2015*; *Murphy et al., 2015*; *Yu et al., 2015*; *Navajas et al., 2016*; *van den Berg et al., 2016*; *Fleming and Daw, 2017*), such as integration of existing information, processing of novel information arriving post-decisionally, or decay (*Moran et al., 2015*), and may consequently be only partly reflective of early confidence-related states.

Here we aimed to derive a more faithful representation of these early confidence signals using EEG, and exploit the trial-by-trial variability in these signals to build parametric EEG-informed fMRI predictors, thus providing a starting point to a more comprehensive spatiotemporal account of decision confidence. We hypothesised that using an electrophysiologically-derived (i.e., endogenous) representation of confidence to detect associated fMRI responses would provide not only a more

temporally precise, but also a more accurate spatial representation of confidence around the time of decision.

To test this hypothesis, we collected simultaneous EEG-fMRI data while participants performed a random-dot direction discrimination task and rated their confidence in each choice. Using a multivariate single-trial classifier to discriminate between High vs. Low confidence trials in the EEG data, we extracted an early, stimulus-independent discriminant component appearing prior to participants' behavioural response. These early representations of confidence correlated across subjects with measures of confidence predicted by an accumulation-to-bound model of decision making. We then used the trial-to-trial variability in the resulting confidence signal as a predictor for the fMRI response, revealing a positive correlation within a region of the VMPFC not commonly associated with confidence for perceptual decisions. Crucially, activation of this region was unique to our EEG-informed fMRI predictor (i.e., additional to those detected with a conventional fMRI regressor, which relied solely on participants' post-decisional confidence reports). Furthermore, a functional connectivity analysis revealed a link between the activation in the VMPFC, and regions of the prefrontal cortex involved in perceptual decision making and metacognition.

## Results

### Behaviour

Subjects (N = 24) performed a speeded perceptual discrimination task whereby they were asked to judge the motion direction of random dot kinematograms (left vs. right), and rate their confidence in each choice on a 9-point scale (*Figure 1A*). Stimulus difficulty (i.e., motion coherence) was held constant across all trials, at individually determined psychophysical thresholds. We found that on average, subjects indicated their direction decision 994 ms (SD = 172 ms) after stimulus onset and performed correctly on 75% (SD = 5.2%) of the trials. In providing behavioural confidence reports, subjects tended to employ the entire rating scale, showing that subjective confidence varied from trial-to-trial despite perceptual evidence remaining constant throughout the task (*Figure 1B*).

As a general measure of validity of subjects' confidence reports, we first examined the relationship with behavioural task performance. Specifically, confidence is largely known to scale positively with decision accuracy and negatively with response time (*Vickers and Packer, 1982*; *Baranski and Petrusic, 1998*), though this relationship is not perfect, and is subject to individual differences (*Baranski and Petrusic, 1994*; *Fleming et al., 2010*; *Fleming and Dolan, 2012*). As expected, we found a positive correlation with accuracy (subject-averaged R = 0.30; one-sample t-test, t(23) = 13.9, p<0.001) (*Figure 1C*), and a negative correlation with response time (subject-averaged R = −0.27; one-sample t-test, t(23) = −7.8, p<0.001) (*Figure 1D*). Thus, subjects' confidence ratings were generally reflective of their performance on the perceptual decision task.

Next, we asked whether the observed variability in subjects' confidence reports could be explained by sustained fluctuations in attention (i.e., spanning multiple trials). We reasoned that decreases in attention may be reflected as serial correlations in confidence ratings across trials. To test this possibility, we performed a serial autocorrelation regression analysis on a single subject basis, which predicted confidence ratings on the current trial from ratings given on the immediately preceding five trials. On average, this model accounted for only a minimal fraction of the variance in confidence ratings (subject-averaged $R^2$ = 0.07). Finally, we sought to rule out the possibility that trial-to-trial variability in confidence could be explained by potential subtle differences in low-level physical properties of the stimulus that may go beyond motion coherence (e.g., location and/or timing of individual dots). To this end, we compared subjects' confidence reports on the two experimental blocks (consisting of identical sequences of random-dot kinematograms), and found no significant correlation between these (subject-averaged R = 0.02, one-sample t-test, p=0.44). Taken together, these results support the hypothesis that subjects' reports reflected internal fluctuations in their sense of confidence, which are largely unaccounted for by external factors.

### EEG-derived measure of confidence

To identify confidence-related signals in the EEG data, we first separated trials into three confidence groups (Low, Medium, and High) on the basis of subjects' confidence ratings. We then conducted a single-trial multivariate classifier analysis (*Parra et al., 2005*; *Sajda et al., 2009*) on the stimulus-

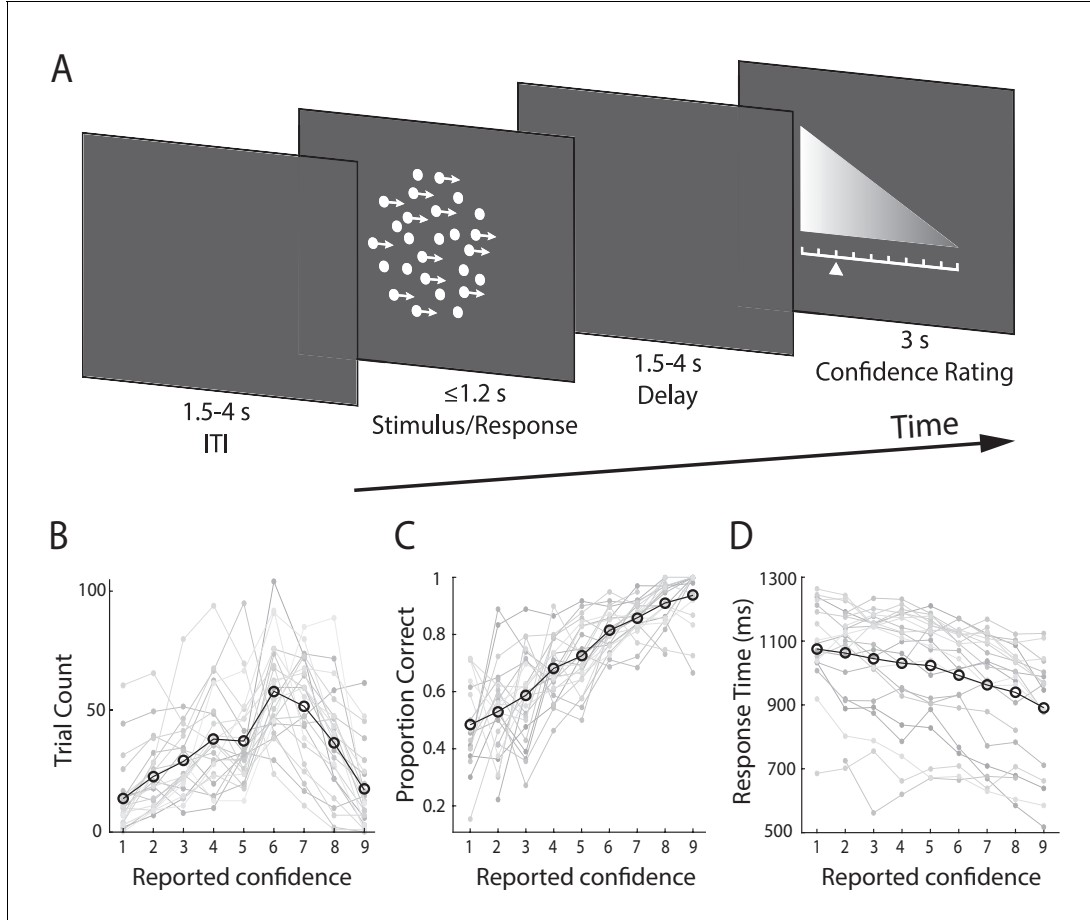

**Figure 1.** Experimental design and behavioural performance. (**A**) Schematic representation of the behavioural paradigm. Subjects made speeded left vs. right motion discriminations of random dot kinematograms calibrated to each individual's perceptual threshold. Stimulus difficulty (i.e., motion coherence) and was held constant across trials. Stimuli were presented for up to 1.2 s, or until a behavioural response was made. After each direction decision, subjects rated their confidence on a 9-point scale (3 s). The response mapping for high vs. low confidence ratings alternated randomly across trials to control for motor preparation effects, and was indicated by the horizontal position of the scale, with the tall end representing high confidence. All behavioural responses were made on a button box, using the right hand. (**B**) Mean confidence rating behaviour, showing the frequency with which subjects selected each point on the confidence scale. (**C**) Mean proportion of correct direction choices as a function of reported confidence. (**D**) Mean response time as a function of reported confidence. Faint grey lines in (**B**), (**C**), and (**D**) indicate individual subject data. For (**C**) and (**D**) we excluded any trial averages based on fewer than five trials.

DOI: https://doi.org/10.7554/eLife.38293.003

locked EEG data, designed to estimate linear spatial weightings of the EEG sensors (i.e., spatial projections) discriminating between Low- vs. High-confidence trials (see Materials and methods). Applying the estimated electrode weights to single-trial data produced a measurement of the discriminating component amplitudes (henceforth $y_{CONF}$), which represent the distance of individual trials from the discriminating hyperplane, and which we treat as a surrogate for the neural confidence of the decision.

Note that even though participants' post-decision ratings may not form an entirely faithful representation of earlier confidence signals, they can nevertheless be used to separate trials into broad confidence groups for training the classifier and estimating the relevant discrimination weights at the time of decision. Data from individual trials, including those not originally used in the discrimination analysis, were subsequently subjected through these electrode weights to obtain a trial-specific graded measure of internal confidence. In other words, these electrophysiologically-derived confidence measures depart from their behavioural counterparts in that they contain trial-to-trial information from the neural generator giving rise to the relevant discriminating components. As such, these

estimates can potentially offer additional insight into the internal processes that underlie confidence at these early stages of the decision.

To quantify the discriminator's performance over time we used the area under a receiver operating characteristic curve (i.e., Az value) with a leave-one-out trial cross validation approach to control for overfitting (see Materials and methods).

We found that discrimination performance (Az) between the two confidence trial groups peaked, on average, 708 ms after stimulus onset (SD = 162 ms, *Figure 2A*; see *Figure 2—figure supplement 1* for Az locked to the time of rating). To visualise the spatial extent of this confidence component, we computed a forward model of the discriminating activity (Materials and methods), which can be represented as a scalp map (*Figure 2A*). Importantly, both the temporal profile and electrode distribution of confidence-related discriminating activity were consistent with our previous work (*Gherman and Philiastides, 2015*) where we used stand-alone EEG to identify time-resolved signatures of confidence during a face vs. car visual categorisation task. Together these observations are an indication that the temporal dynamics of decision confidence can be reliably captured using EEG data acquired inside the MR scanner, and that these early confidence-related signals may generalise across tasks.

To provide additional support linking this discriminating component to choice confidence, we considered the Medium-confidence trials. Importantly, these trials can be regarded as 'unseen' data, as they are independent from those used to train the classifier. We subjected these trials through the same neural generators (i.e., spatial projections) estimated during discrimination of High- vs. Low-confidence trials and, as expected from a graded quantity, found that the mean component amplitudes for Medium-confidence trials were situated between, and significantly different from, those in the High- and Low-confidence trial groups (both p<0.001, *Figure 2B*). To ensure these results were not due to overfitting, we also repeated the above comparisons using fully out-of-sample discriminant component amplitudes obtained from our leave-one-out cross-validation procedure (see Materials and methods), and found that differences remained significant (both p<0.001, *Figure 2—figure supplement 2*)

We next examined the relationship between the confidence-discriminating component and objective performance on the perceptual discrimination task. We found that component amplitudes were positively correlated with decision accuracy (one-sample t-test on logistic regression coefficients, t(23)=8.6, p<0.001, *Figure 2C*), and were consistently higher for correct vs. incorrect responses across subjects (t(23)=7.58, p<0.001, *Figure 2D*), in line with the well-established relationship between confidence and accuracy. To rule out the possibility that the modulation of discriminant component amplitude by confidence was purely explained by objective performance, we compared component amplitudes for Medium-confidence against High-/Low-confidence using only trials associated with correct responses, and showed that differences between these trial groups remained significant (both p<0.001, *Figure 2E*). The same pattern was found when repeating the analysis separately on error trials (both p<0.001). These results indicate that the confidence-related neural component can be dissociated from objective performance, as might be expected from previous reports (*Lau and Passingham, 2006*; *Rounis et al., 2010*; *Komura et al., 2013*; *Lak et al., 2014*; *Fleming and Daw, 2017*).

As the duration of the visual motion stimulus varied across trials in our task (i.e., remained on until subjects made a motor response on the perceptual task) another potential concern might be that the variability in the EEG-derived confidence signatures we identified here could be explained by these stimulus-related factors. We reasoned that if that were the case, we might expect high correlation between stimulus duration and discriminant component amplitudes. However, we found that this correlation was weak (subject-averaged R = -.15), suggesting that our classification results could not have been solely driven by this factor.

Finally, we addressed the possibility that the observed variability in the confidence discriminating component could be attributed to sustained fluctuations in attention, by conducting a serial autocorrelation analysis which predicted component amplitudes on a given trial from those on the preceding five trials (separately for each subject). As before, we expected that if attentional fluctuations are driving the variability in our EEG-derived confidence measures, component amplitudes on a given trial would be reliably predicted by those observed in the immediately preceding trials. We found that this model only explained a small fraction of the variance in component amplitudes (subject-averaged $R^2$ = 0.03).

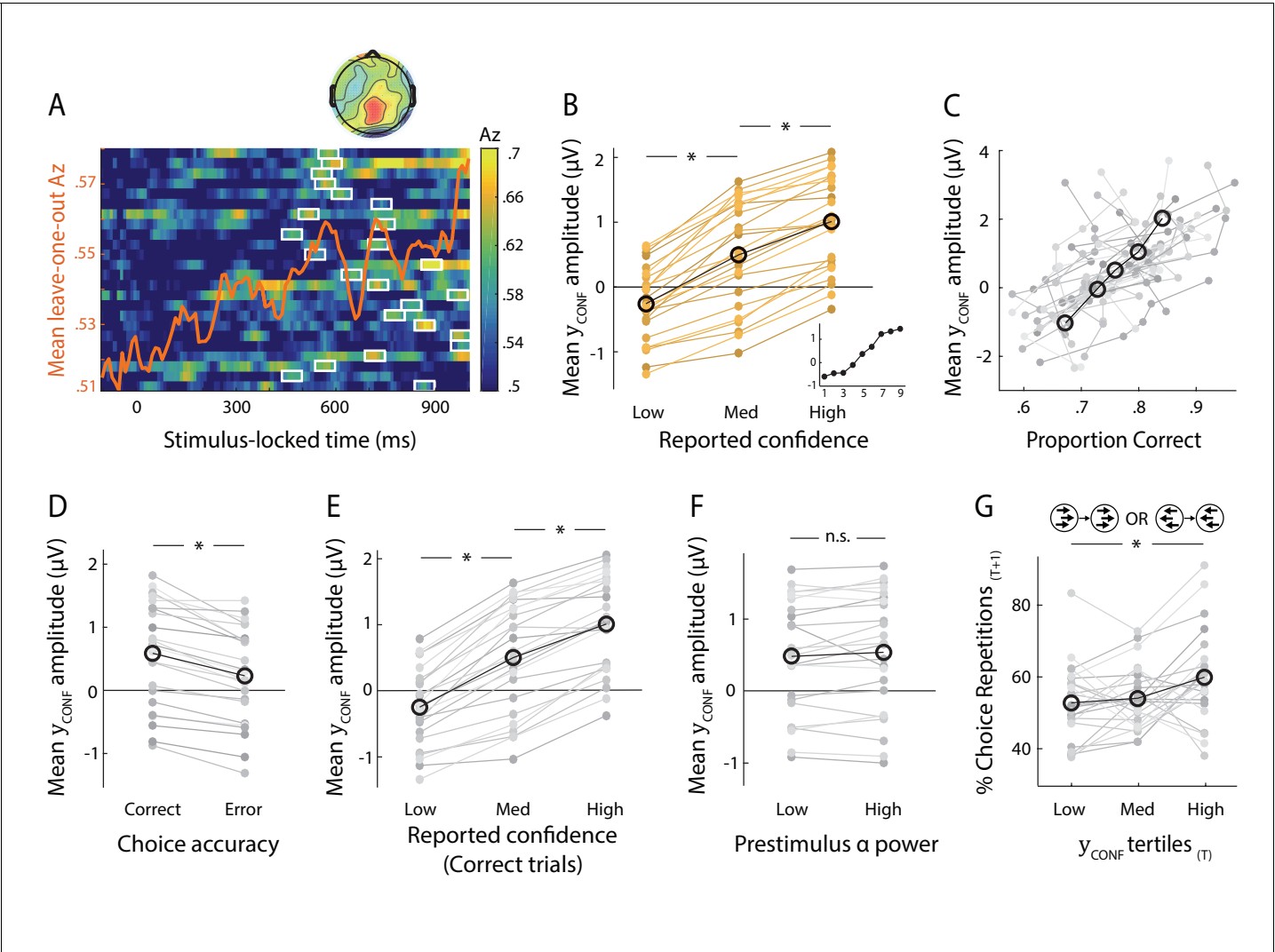

**Figure 2.** Neural representation of confidence in the EEG. (**A**) Classifier performance (Az) during High- vs. Low-confidence discrimination for stimulus-locked data. Each row represents the Az as a function of time, for a single subject (warm colours indicate higher values). The overlapping line (orange) shows the mean classifier performance across subjects. Outlined in white are the pre-response time windows of peak confidence discrimination used subsequently to extract single-trial measures of confidence (i.e., discriminant component amplitudes). In selecting these, we considered only the discrimination period ending, on average, at least 100 ms (across-subject mean 271 ± 162 ms) prior to subjects' mean response times, to minimise potential confounds with activity related to motor execution, due to a sudden increase in corticospinal excitability in this period (**Chen et al., 1998**). Inset shows average (normalised) topography associated with the discriminating component at subject-specific times of peak confidence discrimination. (**B**) Mean amplitude of the confidence discriminant component as a function of reported confidence, showing a parametric effect across the Low, Medium, and High bins. The mean component amplitudes for individual confidence ratings (weighted by each subjects' trial count per rating) are also shown (inset). (**C**) Trial-by-trial confidence discriminant component amplitudes were positively correlated with accuracy. To visualise this relationship, single-trial component amplitudes were grouped into five bins. (**D**) Mean amplitude of the confidence discriminant component for correct vs. error responses, showing a significant effect of choice accuracy.(**E**) Mean amplitude of the confidence discriminant component as a function of reported confidence, for correct trials only (in order to control for accuracy). The same pattern as in (**B**) is observed. (**F**) Mean amplitudes of the confidence discriminant component did not differ significantly between trials associated with High vs. Low prestimulus oscillatory power in the alpha band (which we used as a proxy for subjects' prestimulus attentional state). (**G**) Relationship between the strength of electrophysiological confidence signals on the current trial (i.e., confidence-discriminating component amplitudes) and the tendency to repeat a choice on the immediately subsequent trial, for trial pairs showing stimulus motion in the same direction (i.e., nominally identical stimuli). Faint orange (in **B**) and grey lines (in **C–G**) represent individual subject data.

DOI: https://doi.org/10.7554/eLife.38293.004

The following figure supplements are available for figure 2:

**Figure supplement 1.** Classifier performance (Az) during High- vs Low-confidence discrimination, for data locked to the rating phase of the trial (defined as the onset of the rating prompt).

*Figure 2 continued on next page*

*Figure 2 continued*

DOI: https://doi.org/10.7554/eLife.38293.005

**Figure supplement 2.** Mean amplitude of the confidence discriminant component showing parametric modulation by reported confidence.

DOI: https://doi.org/10.7554/eLife.38293.006

We also assessed the influence of a neural signal known to correlate with attention (*Thut et al., 2006*) and predict visual discrimination (*van Dijk et al., 2008*), namely occipitoparietal prestimulus alpha power. To do this, we separated trials into High vs. Low alpha power groups, individually for each subject, and compared the corresponding average discriminant component amplitudes. We found that these did not differ significantly between the two groups (paired t-test, p=0.19, *Figure 2F*). Note that variability in the confidence discriminant component was also independent of stimulus difficulty, as this was held constant across all trials. In line with this, discriminant component amplitudes for the two identical-stimulus experimental blocks were not significantly correlated (subject-averaged R = 0.02; one-sample t-test, p=0.39).

## Confidence-dependent influences on behaviour

We next sought to identify potential influences of neural confidence signals on decision-related behaviour. In particular, there is evidence that confidence, as reflected in behavioural (*Braun et al., 2018*) and physiological (*Urai et al., 2017*) correlates, can play a role in the modulation of history-dependent choice biases. Here, we tested whether the strength of our EEG-derived confidence signals (i.e., confidence discriminant component amplitude $y_{CONF}$) on a given trial might influence the probability to repeat a choice on the immediately subsequent trial ($P_{REPEAT}$). While we observed no overall significant links between $y_{CONF}$ and subsequent choice behaviour when considering the entire data set, we found a positive relationship between $y_{CONF}$ and $P_{REPEAT}$ *if* stimulus motion on the immediately subsequent trial was in the same direction as in the current trial (F(2,46)=5.89, p=.005, with post-hoc tests showing a significant difference in $P_{REPEAT}$ following Low vs. High $y_{CONF}$ trials, p=.015, Bonferroni corrected), as shown in *Figure 2G*. Thus, stronger confidence signals were associated with an increased tendency to repeat the previous choice.

In contrast, we did not find any modulatory effect of $y_{CONF}$ on choice repetition/alternation behaviour when motion on the current trial was in the opposite direction from that of the previous trial. Thus, choices were only affected by previous confidence when no global change in motion direction had occurred from one trial to the next. Interestingly, this dependence of confidence-related repetition bias on stimulus identity points to a mechanism by which the representation of confidence interacts with a putative process of (subliminal) stimulus-consistency detection (distinguishable from the decision process itself) on the subsequent trial, to influence the decision and/or behaviour.

## Dynamic model of decision making

To seek preliminary insight into how our confidence-related EEG measure relates to the decision formation process, we compared our neural signals with a measure of confidence derived from a dynamic model of decision making. Namely, we fitted subjects' behavioural data (i.e., accuracy and response time) with an adapted version of the race model (*Vickers, 1979*; *Vickers and Packer, 1982*; *De Martino et al., 2013*) (see Materials and methods). This class of models describes the decision process as a stochastic accumulation of perceptual evidence over time by independent signals representing the possible choices (*Figure 3A*). The decision terminates when one of the accumulators reaches a fixed threshold, with choice being determined by the winning accumulator. Importantly, confidence for binary choices can be estimated in these models as the absolute distance ($\Delta e$) between the states of the two accumulators at the time of decision (i.e., 'balance of evidence' hypothesis).

Overall, we found that this model provided a good fit to the behavioural data (Accuracy: R = 0.76, p<0.001, *Figure 3B*; RT: subject-averaged R = 0.965, all p<=0.0016, see *Figure 3—figure supplement 1* for individual subject fits). We illustrate model fits to response time data in *Figure 3C* (see *Figure 3—figure supplement 2* for individual subject fits), whereby response time distributions

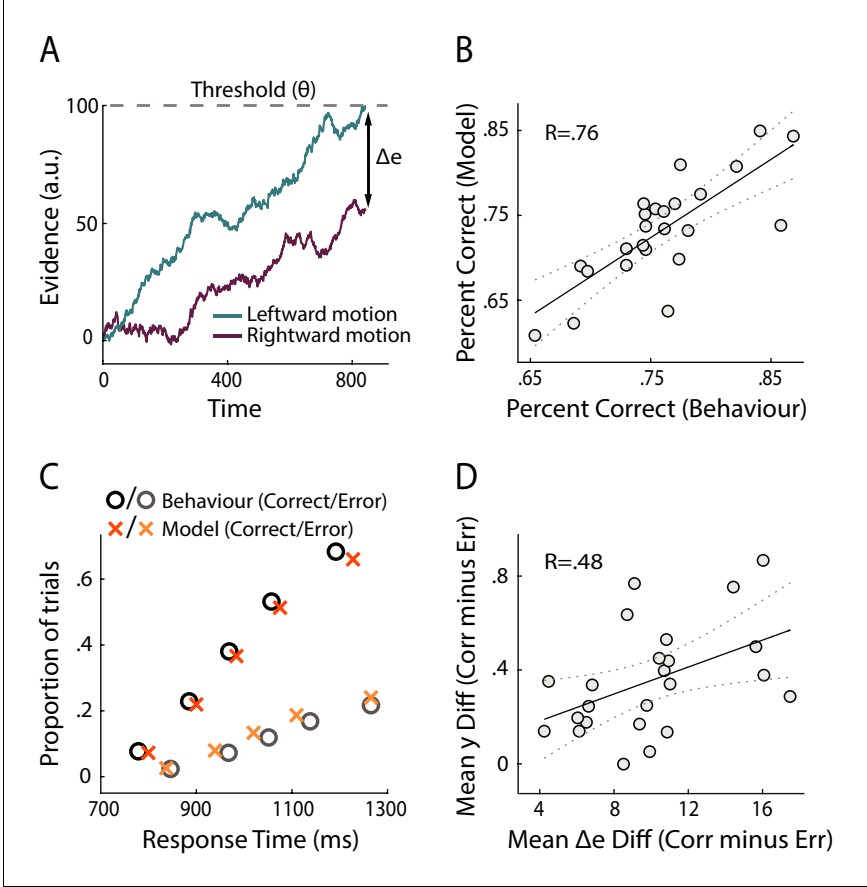

**Figure 3.** Modelling results. (**A**) Schematic representation of the decision model for one trial. Evidence in favour of the two choice alternatives (here, leftward and rightward motion) accumulates gradually over time. A decision is made when one of the accumulators reaches a decision threshold (θ). The model quantifies confidence as the absolute difference in the accumulated evidence for the two options, at the time of decision (Δe). (**B**) Correlation between behavioural vs. model-predicted choice accuracy. Each point represents trial-averaged data for one subject. (**C**) Behavioural (circles) and model-predicted (crosses) response time distribution. On the x axis from left to right, data points represent the RT below which 10%, 30%, 50%, 70% and 90% of the data, respectively, are situated. The y axis shows the associated proportion of data for correct (upper symbols) and incorrect (bottom symbols) responses. (**D**) Across-subject correlation between the model-predicted and neurally observed relationship of confidence with choice accuracy (quantified as the difference in confidence estimates between correct and error trials). Each dot represents data for one subject.

DOI: https://doi.org/10.7554/eLife.38293.007

The following figure supplements are available for figure 3:

**Figure supplement 1.** Model fits for individual subjects.

DOI: https://doi.org/10.7554/eLife.38293.008

**Figure supplement 2.** Behavioural (circles) and model-predicted (crosses) response time distribution for individual subjects.

DOI: https://doi.org/10.7554/eLife.38293.009

for correct and error trials are summarised separately using five quantile estimates of the associated cumulative distribution functions (*Forstmann et al., 2008*).

Here, we were interested in how our neural measures of confidence (EEG-derived discriminant component $y_{CONF}$) compared against the confidence estimates predicted by the decision model (Δe), at the subject group level. To this end, we computed the mean difference in confidence (as reflected by $y_{CONF}$ and Δe, respectively) between correct and error trials, separately for each subject, and tested the extent to which these quantities were correlated across participants. This relative measure, which captured the relationship between confidence and choice accuracy, also ensured

that comparisons across subjects remained meaningful after averaging across trials. We found a significant positive correlation (i.e., subjects who showed stronger difference in $y_{CONF}$ between correct and error trials also showed a higher difference in $\Delta e$, R=.48, p=.019, robust correlation coefficient obtained using the percentage bend correlation analysis (Wilcox, 1994; see *Figure 3D*), opening the possibility that neural confidence signals might be informed by a process similar to the race-like dynamic implemented by the current model.

## Exploratory mediation analysis

We sought to further clarify the link between model-derived confidence estimates ($\Delta e$), early neural signatures of confidence ($y_{CONF}$), and subjects' behavioural reports during the rating phase of the trial (Ratings), by performing an exploratory mediation analysis on these measures. We hypothesised that $y_{CONF}$ may be informed by quantities equivalent to $\Delta e$, and in turn influence the confidence estimates reflected in post-choice reports. Thus, we tested whether $y_{CONF}$ may act as a statistical mediator on the link between $\Delta e$ and Ratings. As with our previous analysis linking $y_{CONF}$ and $\Delta e$ (*Figure 3D*), we first computed the mean difference between correct and error trials for each of the three variables of interest, to produce comparable measures across subjects (i.e., by removing potentially task-irrelevant individual differences in the trial-averaged scores, such as rating biases). These quantities (henceforth referred to as $\Delta e_{DIFF}$, $y_{CONF\_DIFF}$, and Ratings$_{DIFF}$) were then submitted to the mediation analysis.

Specifically, we defined a three-variable path model (Wager et al., 2008) with $\Delta e_{DIFF}$ as the predictor variable, Ratings$_{DIFF}$ as the dependent variable, and $y_{CONF\_DIFF}$ as the mediator (Materials and methods). In line with our prediction, we found that: 1) $\Delta e_{DIFF}$ was a significant predictor of $y_{CONF\_DIFF}$ (p=.01), 2) $y_{CONF\_DIFF}$ reliably predicted Ratings$_{DIFF}$ after accounting for the effect of predictor $\Delta e_{DIFF}$ (p<.001), and 3) the indirect effect of $y_{CONF\_DIFF}$, defined as the coefficient product of effects 1) and 2), was also significant (p=.004). While the across-subject nature of the analysis calls for caution in interpreting the results, these observations are consistent with the possibility that $y_{CONF}$ reflects a (potentially noisy) readout of decision-related balance of evidence (as modelled by $\Delta e$), and informs eventual confidence reports.

## fMRI correlates of confidence

We sought primarily to identify fMRI activations correlating uniquely with the endogenous signatures of confidence at the time of the perceptual decision, as obtained from our EEG discrimination analysis. In particular, we were interested in confidence-related variability in the fMRI response that might be over and above what can be inferred from behavioural confidence reports alone. To this end, we constructed a general linear model (GLM; see Materials and methods) of the fMRI using an EEG-derived regressor for confidence ($y_{CONF}$) together with additional regressors accounting for variance related to subjects' behavioural confidence reports (i.e., ratings), and other potentially confounding factors (task performance, response time, attention, and visual stimulation).

*fMRI correlates of behavioural confidence reports*. We first investigated the activation patterns associated with confidence ratings during the perceptual decision phase of the trial (*Figure 4A*), defined as the time window beginning at the onset of the random-dot stimulus (and ending prior to the onset of the confidence rating prompt). The coordinates of all activations are listed in Supplementary Table 1 (*Supplementary file 1*). We found that the BOLD response increased with reported confidence in the striatum, lateral orbitofrontal cortex (OFC), the ventral anterior cingulate cortex (ACC) – areas thought to play a role in human valuation and reward (*O'Doherty, 2004*; *Rushworth et al., 2007*; *Grabenhorst and Rolls, 2011*) – as well as the right anterior middle frontal gyrus, amygdala/hippocampus, and visual association areas. Overall, these activations appear consistent with findings from previous studies that have identified spatial correlates of decision confidence (*Rolls et al., 2010*; *De Martino et al., 2013*; *Heereman et al., 2015*; *Hebart et al., 2016*). Negative activations (i.e., regions showing increasing BOLD response with decreasing reported confidence) were found in the right supplementary motor area, dorsomedial prefrontal cortex, right inferior frontal gyrus (IFG), anterior insula/frontal operculum, in line with previous reports of decision uncertainty near the time of decision (*Heereman et al., 2015*; *Hebart et al., 2016* ).

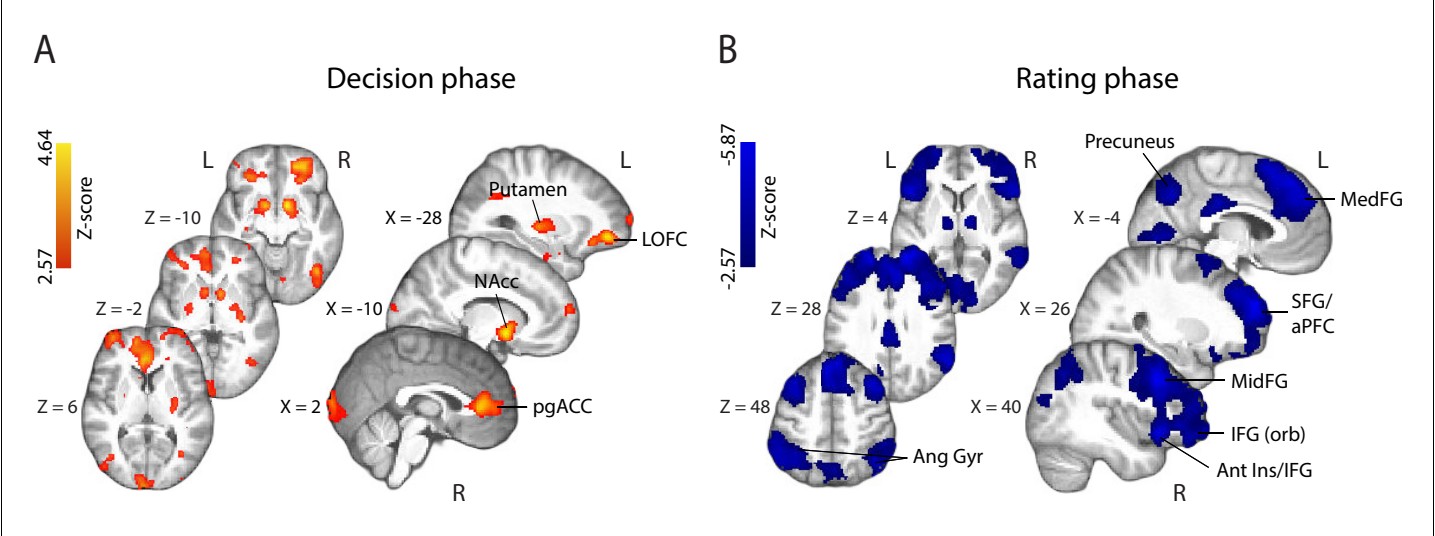

**Figure 4.** Parametric modulation of the BOLD signal by reported confidence. (A) Clusters showing positive correlation with confidence during the decision phase of the trial. (B) Clusters showing negative correlation with confidence at the onset of the rating cue (i.e., rating phase). All results are reported at |Z| ≥ 2.57, and cluster-corrected using a resampling procedure (minimum cluster size 162 voxels; see Materials and methods). *Ang Gyr*, angular gyrus; *Ant Ins*, anterior insula; *IFG (orb)*, inferior frontal gyrus (orbital region); *LOFC*, lateral orbitofrontal cortex; *MedFG*, medial frontal gyrus; *MidFG*, middle frontal gyrus; *NAcc*, nucleus accumbens; *pgACC*, pregenual anterior cingulate cortex; *RLPFC*, rostrolateral prefrontal cortex; *SFG*, superior frontal gyrus. The complete lists of activations are shown in Supplementary Tables 1 and 2 (*Supplementary file 1*).
DOI: https://doi.org/10.7554/eLife.38293.010

During the metacognitive report stage of the trial (i.e., 'rating phase', defined as the time window beginning at the onset of the confidence prompt; *Figure 4B*), we found negative correlations with confidence ratings in extended networks (Supplementary Table 2; *Supplementary file 1*) which included regions of the rostrolateral prefrontal cortex (bilateral, right lateralised), middle frontal gyrus, superior frontal gyrus (extending along the cortical midline and into the medial prefrontal cortex), orbital regions of the IFG, angular gyrus, precuneus, posterior cingulate cortex (PCC), and regions of the occipital and middle temporal cortices. These activations are largely in line with research on the spatial correlates of choice uncertainty (*Grinband et al., 2006*; *Fleming et al., 2012*; ) and metacognitive evaluation (*Fleming et al., 2010*; *Molenberghs et al., 2016*). Finally, positive correlations were observed in the striatum and amygdala/hippocampus, as well as motor cortices.

*fMRI correlates of EEG-derived confidence signals.* To identify potential brain regions encoding early representations of confidence as captured by our confidence-discriminating EEG component, we turned to the parametric EEG-derived fMRI regressor (i.e., $y_{CONF}$ regressor), which captured the inherent single-trial variability in these signals. Our approach therefore allowed us to model the fMRI response using time-resolved neural signatures of confidence, which were specific to each subject. Crucially, as these measures captured the variability in the neural representation of confidence near the time of the perceptual decision itself (i.e., prior to behavioural response), they may be better suited for spatially characterising confidence during this time window compared to the behavioural confidence reports obtained later on in the trial (as the latter may be more reflective of confidence-related information arriving post-decisionally). Note that these signals were only moderately correlated with reported confidence (subject-averaged R=.39, SD=.07), and thus could potentially provide additional explanatory power in our fMRI model.

This EEG-informed fMRI analysis revealed a large cluster in the ventromedial prefrontal cortex (VMPFC, peak MNI coordinates [−8 40 − 14]), extending into the subcallosal region and ventral striatum, and a smaller cluster in the right precentral gyrus (peak MNI coordinates [30 -20 64]), where the BOLD response correlated positively with the EEG-derived confidence discriminating component (*Figure 5*). The VMPFC has been linked to confidence-related processes in value-based, as well as other complex decisions (*De Martino et al., 2013*; *Lebreton et al., 2015*), however this region is

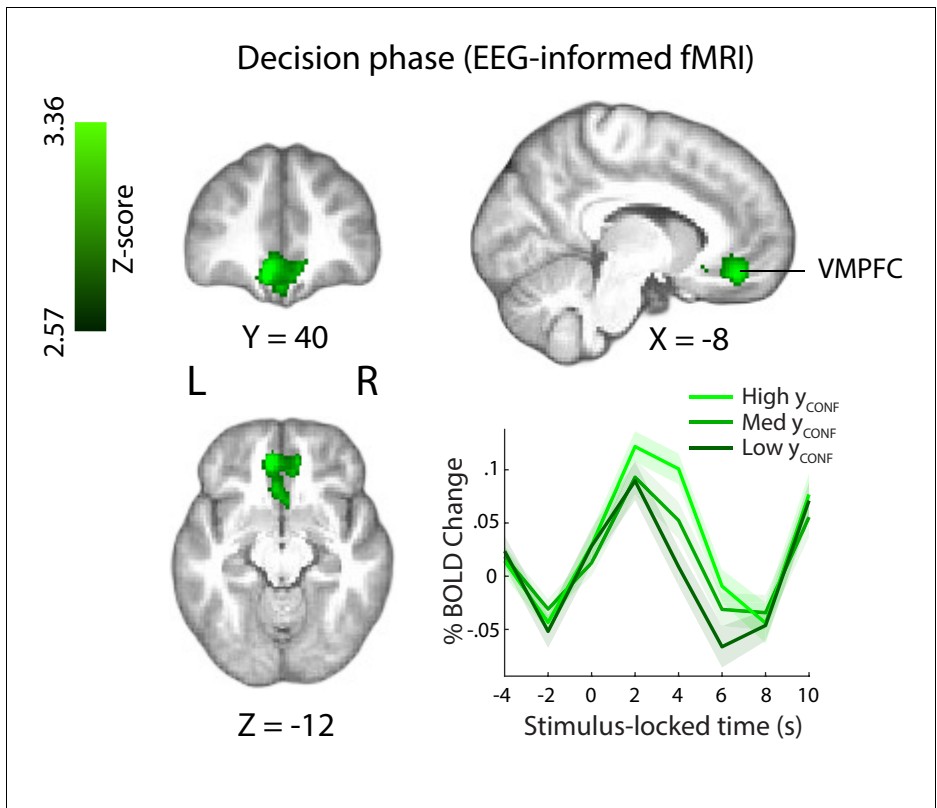

**Figure 5.** Positive parametric modulation of the BOLD signal by an EEG-derived single-trial confidence measure (see Materials and methods), during the decision phase of the trial. Results are reported at $|Z| \geq 2.57$, and cluster-corrected using a resampling procedure (minimum cluster size 162 voxels). *Bottom right:* Time course of VMPFC BOLD response, showing parametric modulation by neural confidence (presented for illustration purposes only). Trials are separated by the strength of confidence-discriminating component amplitudes ($y_{CONF}$). *VMPFC,* ventromedial prefrontal cortex.

DOI: https://doi.org/10.7554/eLife.38293.011

The following figure supplements are available for figure 5:

**Figure supplement 1.** Positive correlation of the BOLD signal with the EEG-derived confidence measures in the posterior cingulate cortex (PCC), during the decision phase of the trial.

DOI: https://doi.org/10.7554/eLife.38293.012

**Figure supplement 2.** Positive parametric modulation of the BOLD signal by EEG-derived confidence at the confidence rating stage.

DOI: https://doi.org/10.7554/eLife.38293.013

**Figure supplement 3.** Correlations between HRF-convolved regressors locked to stimulus (i.e., decision phase) and confidence rating prompt (i.e., rating phase).

DOI: https://doi.org/10.7554/eLife.38293.014

**Figure supplement 4.** Parametric modulation of the BOLD signal by confidence, resulting from two GLM analyses whereby events pertaining to the decision and rating phases of the trial, respectively, were modelled separately.

DOI: https://doi.org/10.7554/eLife.38293.015

**Figure supplement 5.** Positive parametric modulation of the BOLD signal by EEG-derived measures of confidence resulting from a leave-one-trial-out cross validation procedure (shown in pink).

DOI: https://doi.org/10.7554/eLife.38293.016

not typically associated with confidence in perceptual decisions (though see *Heereman et al., 2015*; *Fleming et al., 2018*).

Note also that, as regression parameter estimates resulting from standard GLM analysis reflect variability unique to each regressor (i.e., disregarding common variability) (Mumford et al. 2015), the correlation we observed with the EEG-derived $y_{CONF}$ regressor in the VMPFC during the perceptual decision period is over and above what can be explained by behavioural confidence ratings alone (i.

e., the Ratings$_{DEC}$ regressor, *Figure 4A*). Consistent with this, correlation of the Ratings$_{DEC}$ regressor with activity in the relevant VMPFC cluster (including in a supplementary GLM analysis whereby the $y_{CONF}$ regressor was removed) failed to pass statistical thresholding and would have therefore been missed using behavioural ratings alone.

Interestingly, the scalp map associated with our confidence discriminating EEG component showed a diffused topography including contributions from several centroparietal electrode sites. One possibility is that the observed spatial pattern reflects sources of shared variance between the EEG component and confidence ratings themselves (which was otherwise controlled for in our original fMRI analysis). To test this, we ran a separate control GLM analysis where the confidence ratings regressor (Ratings$_{DEC}$) was removed, and found that with this model the $y_{CONF}$ regressor explained additional variability of the BOLD signal within several regions, including precuneus/PCC regions of the parietal cortex (*Figure 5—figure supplement 1*). Notably, activity in these regions has been previously shown to scale with confidence (*De Martino et al., 2013*; *White et al., 2014*) and hypothesised to play a role in metacognition (*McCurdy et al., 2013*).

In a separate analysis, we also explored BOLD signal correlations with the $y_{CONF}$ regressor locked to the confidence rating stage (as part of a GLM model which only included regressors at the time of rating). We found no correlation with $y_{CONF}$ in the VMPFC, suggesting confidence-related activation in this region was specific to the earlier stages of the decision. Clusters showing positive correlation with $y_{CONF}$ were found in the (bilateral) motor cortex, left planum temporale, putamen/pallidum, and lateral occipital cortex (*Figure 5—figure supplement 2*). Suggestive mainly of motor-related processes, these activations may have been partially confounded by repeated movement (i.e., button pushes) during the rating stage of the trial. More speculatively, confidence representations may be present within motor regions, in line with the idea that decision-related information 'leaks' into the motor systems that support relevant action (*Gold and Shadlen, 2000*; *Song and Nakayama, 2009*). We found no clusters showing negative correlation with $y_{CONF}$ at this stage of the trial.

## Psychophysiological interaction (PPI) analysis

Having identified the VMPFC as uniquely encoding a confidence signal early on in the trial (i.e., near the time of the perceptual decision), we next sought to explore potential functional interactions of this region with the rest of the brain (for instance, with networks involved in perceptual decision making and/or post-decision metacognitive processes). To this end, we conducted a whole-brain PPI analysis (see Materials and methods), whereby we searched for areas showing increased correlation of their BOLD response with that of a VMPFC seed, during the perceptual decision phase of the trial (i.e., defined here as the trial-by-trial time window between the onset of the motion stimulus and subject's explicit commitment to choice).

Based on existing literature showing negative BOLD correlations with confidence ratings in regions recruited post-decisionally (e.g., during explicit metacognitive report), such as the anterior prefrontal cortex (*Fleming et al., 2012*; *Hilgenstock et al., 2014*; *Morales et al., 2018*), we expected that increased functional connectivity of such regions with the VMPFC would be reflected in stronger negative correlation in our PPI. Similarly, we hypothesised that fMRI activity in regions encoding the perceptual decision would also correlate negatively with confidence/VMPFC activation, in line with the idea that easier (and thus more confident) decisions are characterised by faster evidence accumulation to threshold (*Shadlen and Newsome, 2001*) and weaker fMRI signal in reaction time tasks (*Ho et al., 2009*; *Kayser et al., 2010*; *Liu and Pleskac, 2011*; *Filimon et al., 2013*; *Pisauro et al., 2017*). Accordingly, we expected that if such regions increased their functional connectivity with the VMPFC during the decision, this would manifest as stronger negative correlation in the PPI analysis.

We found that clusters in the bilateral orbitofrontal cortex (OFC; peak MNI: [16 18 -16] and [−28 28–20]), left anterior prefrontal cortex (aPFC; peak MNI: [−40 46 4]), and right dorsolateral prefrontal cortex (dlPFC; peak MNI: [48 22 30]) (*Figure 6*) showed increased negative correlation with VMPFC activation during the perceptual decision. Interestingly, regions in the aPFC and dlPFC in particular have been previously linked to perceptual decision making (*Noppeney et al., 2010*; *Liu and Pleskac, 2011*; *Philiastides et al., 2011*; *Filimon et al., 2013*), as well as post-decisional confidence-related processes (*Fleming et al., 2012*; *Hilgenstock et al., 2014*; *Morales et al., 2018*) and metacognition (*Fleming et al., 2010*; *Rounis et al., 2010*; *McCurdy et al., 2013*).

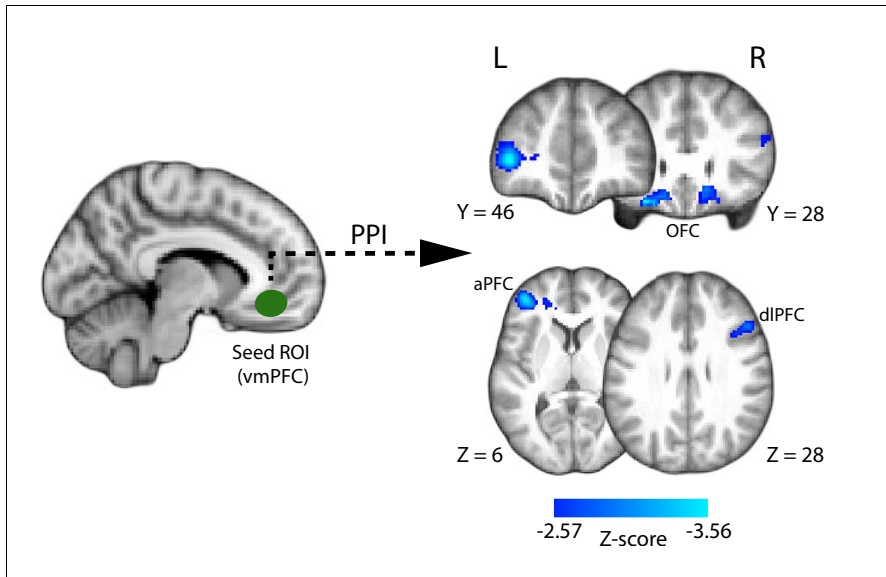

**Figure 6.** Psychophysiological interaction (PPI) analysis showing functional connectivity with the ventromedial prefrontal cortex (i.e., the seed region of interest; approximate location shown in green) during the perceptual decision phase of the trial. Clusters in the anterior and dorsolateral prefrontal cortices, as well as the orbitofrontal cortex (shown in blue), show increased negative correlation with the VMPFC during the perceptual decision. All results are reported at |Z| ≥ 2.57, and cluster-corrected using a resampling procedure (minimum cluster size 162 voxels).

DOI: https://doi.org/10.7554/eLife.38293.017

## Discussion

Here, we used a simultaneous EEG-fMRI approach to investigate the neural correlates of confidence during perceptual decisions. Our method capitalised on the unique explanatory power of time-resolved, internal measures of confidence to identify associated responses in the fMRI, allowing for a more precise spatiotemporal characterisation of confidence than if relying solely on behavioural measures. We found that BOLD response in the VMPFC was uniquely explained by the single-trial variability in an early, EEG-derived neural signature of confidence occurring prior to subjects' behavioural expression of response. This activity was additional to what could be explained by subjects' behavioural reports alone. Our results provide empirical support for the involvement of the VMPFC in confidence of perceptual decisions, and suggest that this region may support an early readout of confidence (i.e., at, or near, the time of decision) preceding explicit choice or metacognitive evaluation.

We first showed that our EEG results - namely the temporal and spatial profile of the confidence-discriminating activity - were consistent with our previous work (*Gherman and Philiastides, 2015*) where we used a different perceptual task involving face vs. car visual categorisations, indicating that these confidence-related signals may generalise across a broader range of tasks. Interestingly, the spatial topography associated with this activity appears consistent with centroparietal scalp projections arising from signals culminating near the decision (*O'Connell et al., 2012*; *Kelly and O'Connell, 2013*; *Philiastides et al., 2014*). While the spatial limitation of EEG precludes conclusive interpretations based on this similarity, this pattern could potentially reflect a mixture of decision- and confidence-related signals, in line with the evidence that suggests these quantities may unfold together around the decision process itself (*Kiani and Shadlen, 2009*; *Gherman and Philiastides, 2015*; *van den Berg et al., 2016*; *Dotan et al., 2018*). Signals such as the centroparietal positivity (CPP) (*O'Connell et al., 2012*) and/or related P300 may themselves hold information about confidence as suggested by electrophysiological work (*Boldt and Yeung, 2015*) (see also (*Urai and Pfeffer, 2014*; *Twomey et al., 2015*) for brief discussions).

Further, our fMRI data revealed activation patterns suggesting that distinct neural networks carry information about confidence during perceptual decision vs. explicit confidence reporting stages of

the trial, respectively. Indeed, it seems plausible that qualitatively distinct representations of confidence may be encoded at different times relative to the decision process. In particular, activations during the decision phase of the trial such as the VMPFC or anterior cingulate cortex, are in line with a more automatic encoding of confidence, i.e., in the absence of explicit confidence report (*Lebreton et al., 2015*; *Bang and Fleming, 2018*). In line with this idea, we also observed activations in regions associated with the human reward/valuation system, such as the striatum and orbitofrontal cortex. In contrast, regions showing correlation with confidence during the confidence rating stage, in particular the anterior prefrontal cortex, have been previously associated with explicit metacognitive judgment/report (*Fleming et al., 2012*; *Morales et al., 2018*), potentially serving a role in higher-order monitoring and confidence communication.

We presented several findings that sought to further clarify the nature and role of the early confidence signals observed in the EEG data, as well as their relationship with the perceptual decision and metacognition. Our computational modelling approach provided preliminary insight into the potential decision dynamics that might inform early confidence. Namely, we showed that these neural signals were consistent with predictions from a dynamic model of decision that quantifies confidence as the difference in accumulated evidence in favour of the possible choice alternatives, at the termination of the decision process. A possible interpretation is that the early confidence representations reflect a readout of this difference (for instance, by a distinct system than the one supporting the perceptual choice itself). In other words, early confidence representations could be informed by, yet be distinct from, the quantities reflected in the model-derived confidence, in line with a dissociation between the information supporting the decision vs. confidence. Our exploratory mediation analysis is in agreement with this interpretation, suggesting that EEG-derived confidence representations can be thought of as a statistical mediator between model-derived confidence measures (reflecting the balance of accumulated evidence at the time of decision) and confidence ratings.

In another exploratory analysis that aimed to better understand the potential impact of neural confidence signals on subsequent behaviour, we found that stronger signal amplitude increased the likelihood of repeating a choice on the subsequent trial, when the motion direction of the stimulus was consistent with that of the previous trial. Interestingly however, we did not observe this effect when subsequent motion was in the opposite direction. This dependence of the confidence-related choice repetition bias on stimulus identity is counterintuitive yet intriguing, as it points to a process that detects stimulus consistency (i.e., independently of the decision process itself), which interacts with representations of previous confidence to alter decision/behaviour (e.g., through selective re-weighting of evidence). While our current decision model cannot account for this confidence-driven trial-to-trial dependence, future computational developments may help reconcile these observations with formal models of decision and confidence.

Our main fMRI finding, linking early confidence representations with VMPFC activity suggests partial independence of these signals from decision centres. Specifically, as the VMPFC is not typically known to support perceptual decision processes, it seems more plausible that the confidence signals we observe here represent a (potentially noisy) readout of confidence-related information. In line with this, computational and neurobiological accounts of confidence processing have proposed architectures by which a first-level form of confidence in a decision emerges as a natural property of the neural processes that support the decision, and in turn is read out (i.e., summarised) by separate higher-order monitoring network(s) (*Insabato et al., 2010*; *Meyniel et al., 2015*; *Pouget et al., 2016*).

The timing of our EEG-derived confidence representations arising in close temporal proximity to the decision (but prior to commitment to a motor response) further endorse the hypothesis that the VMPFC may encode an automatic readout of confidence (*Lebreton et al., 2015*) in decision making, or early (and automatic) 'feeling of rightness' (*Hebscher and Gilboa, 2016*) in memory judgments. While dedicated research will be necessary to establish the functional role of these early signals, fast pre-response confidence signals could be necessary to regulate the link between decision and impending action, for example with low confidence signalling the need for additional evidence (*Desender et al., 2018*).

Consistent with a role in providing a confidence readout, recent work suggests the VMPFC may encode confidence in a task-independent and possibly domain-general manner. Specifically, several functional neuroimaging studies have shown positive modulation of VMPFC activation by confidence, across a range of decision making tasks (*Rolls et al., 2010*; *De Martino et al., 2013*;

*Heereman et al., 2015*; *Lebreton et al., 2015*; *Fleming et al., 2018*). Notably, one study showed that fMRI activation in the VMPFC was modulated by confidence across four different tasks involving both value-based and non-value based rating judgments (*Lebreton et al., 2015*). Furthermore, evidence from memory-related decision making research appears to also implicate the VMPFC in confidence processing (*Hebscher and Gilboa, 2016*).

An outstanding question is whether, and how, the early confidence signals we identified in the VMPFC might further contribute to post-decisional metacognitive signals and eventual confidence reports. It has been long proposed that metacognitive evaluation relies on additional processes taking place post-decisionally (*Pleskac and Busemeyer, 2010*; *Moran et al., 2015*; *Yu et al., 2015*). For instance, recent evidence suggests that choice itself (and corresponding motor-related activity) affects confidence (*Fleming et al., 2015*; *Gajdos et al., 2018*) and may help calibrate metacognitive reports (*Siedlecka et al., 2016*; *Fleming and Daw, 2017*). The early confidence signals in the VMPFC could serve as one of multiple inputs to networks supporting retrospective metacognitive processes, e.g., anterior prefrontal regions (*Fleming et al., 2012*). Interestingly, our functional connectivity analysis revealed a strengthening of the link between the VMPFC and frontal areas (notably the aPFC and dlPFC) during the perceptual decision stage of the trial. While the functional significance of these connections remains to be determined, previous involvement of these regions in perceptual decision making and metacognition makes them likely candidates for providing or receiving input to/from the VMPFC within a confidence-related network.

The observation that the VMPFC, a region known for its involvement in choice-related subjective valuation (*Philiastides et al., 2010*; *Rangel and Hare, 2010*; *Bartra et al., 2013*; *Pisauro et al., 2017*) encodes confidence signals during perceptual decisions raises an interesting possibility for interpreting our results. Our behavioural paradigm did not involve any explicit reward/feedback manipulation and accordingly, the observed confidence-related activation cannot be interpreted as an externally driven value signal. Instead, as has been suggested previously (*Barron et al., 2015*; *Lebreton et al., 2015*), a likely explanation is that as an internal measure of performance accuracy, confidence is inherently valuable. Such a signal may represent *implicit* reward and possibly act as a teaching signal (*Daniel and Pollmann, 2012*; *Guggenmos et al., 2016*; *Hebart et al., 2016*; *Lak et al., 2017*) to drive learning.

In line with this interpretation, recent work suggests that confidence may be used in the computation of prediction errors (i.e., the difference between expected and currently experienced reward) (*Lak et al., 2017*; *Colizoli et al., 2018*), thus guiding a reinforcement-based learning mechanism. Relatedly, confidence prediction error (the difference between expected and experienced confidence) has been hypothesised to act as a teaching signal and guide learning in the absence of feedback. In particular, regions in the human mesolimbic dopamine system, namely the striatum and ventral tegmental area, have been shown to encode both anticipation and prediction error related to decision confidence, in the absence of feedback (*Guggenmos et al., 2016*), similarly to what is typically observed during reinforcement learning tasks where feedback is explicit (*Preuschoff et al., 2006*; *Fouragnan et al., 2015*; *Fouragnan et al., 2017*; *Fouragnan et al., 2018*). Importantly, these effects were predictive of subjects' perceptual learning efficiency. Thus, confidence in valuation/ reward networks could be propagated back to the decision systems to optimize the dynamics of the decision process, possibly by means of a reinforcement-learning mechanism. At the neural level, this could be implemented through a mechanism of strengthening or weakening information processing pathways that result in high and low confidence, respectively (*Guggenmos and Sterzer, 2017*). Though testing this hypothesis extends beyond the scope of the current study, we might expect that fluctuations in expected vs. actual confidence signals observed in our data have a similar influence on learning (e.g., perceptual learning (*Law and Gold, 2009*; *Kahnt et al., 2011*; *Diaz et al., 2017*).

In conclusion, we showed that by employing a simultaneous EEG-fMRI approach, we were able to localise an early representation of confidence in the brain with higher spatiotemporal precision than allowed by fMRI alone. In doing so, we provided novel empirical evidence for the encoding of a generalised confidence readout signal in the VMPFC preceding explicit metacognitive report. Our findings provide a starting point for further investigations into the neural dynamics of confidence formation in the human brain and its interaction with other cognitive processes such as learning, and the decision itself.

## Materials and methods

### Participants

Thirty subjects participated in the simultaneous EEG-fMRI experiment. Four were subsequently removed from the analysis due to near chance (n = 3) and near ceiling (n = 1) performance, respectively, on the perceptual discrimination task. Additionally, one subject was excluded whose confidence reports covered only a limited fraction of the provided rating scale, thus yielding an insufficient number of trials to be used in the EEG discrimination analysis (see below). Finally, one subject had to be removed due to poor (chance) performance of the EEG decoder. All results presented here are based on the remaining 24 subjects (age range 20 – 32 years). All were right-handed, had normal or corrected to normal vision, and reported no history of neurological problems. The study was approved by the College of Science and Engineering Ethics Committee at the University of Glasgow (CSE01355) and informed consent was obtained from all participants. While we conducted no explicit power analysis for determining sample size, note that our EEG analysis was performed on individual subjects using cross validation, such that in estimating our electrophysiologically-derived measure of confidence, each subject became their own replication unit (*Smith and Little, 2018*).

### Stimuli and task

All stimuli were created and presented using the PsychoPy software (*Peirce, 2007*). They were displayed via an LCD projector (frame rate = 60 Hz) on a screen placed at the rear opening of the bore of the MRI scanner, and viewed through a mirror mounted on the head coil (distance to screen = 95 cm). Stimuli consisted of random dot kinematograms (*Newsome and Pare, 1988*), whereby a proportion of the dots moved coherently to one direction (left vs. right), while the remainder of the dots moved at random. Specifically, each stimulus consisted of a dynamic field of white dots (number of dots = 150; dot diameter = 0.1 degrees of visual angle, dva; dot life time = 4 frames; dot speed = 6 dva/s), displayed centrally on a grey background through a circular aperture (diameter = 6 dva). Task difficulty was controlled by manipulating the proportion of dots moving coherently in the same direction (i.e., motion coherence).

We aimed to maintain overall performance on the main perceptual decision task consistent across subjects (i.e., near perceptual threshold, at approximately 75% correct). For this reason, task difficulty was calibrated individually for each subject on the basis of a separate training session, prior to the day of the main experiment.

#### Training

To first familiarise subjects with the random dot stimuli and facilitate learning on the motion discrimination task, subjects first performed a short simplified version of the main task (lasting approx. 10 min), where feedback was provided on each trial. The task, which required making speeded direction discriminations of random dot stimuli (see below), began at a low-difficulty level (motion coherence = 40%) and gradually increased in difficulty in accordance with subjects' online behavioural performance (a 3-down-1-up staircase procedure, where three consecutive correct responses resulted in a 5% decrease in motion coherence, whereas one incorrect response yielded a 5% increase). This was followed by a second, similar task, which served to determine subject-specific psychophysical thresholds. Seven motion coherence levels (5%, 8%, 12%, 18%, 28%, 44%, 70%) were equally and randomly distributed across 350 trials. The proportion of correct responses was separately computed for each motion coherence level, and a logarithmic function was fitted through the resulting values in order to estimate an optimal motion coherence yielding a mean performance of approximately 75% correct. Subjects who showed near-chance performance across all coherence levels or showed no improvement in performance with increasing motion coherence were not tested further and did not participate in the main experiment. No feedback was given for this or any of the subsequent tasks.

#### Main task

On the day of the main experiment, subjects practised the main task once outside the scanner, and again inside the scanner prior to the start of the scan (a short 80 trial block each time). Subjects

made left vs. right direction discriminations of random dot kinematograms and rated how confident they were in their choices, on a trial-by-trial basis (*Figure 1A*). Each trial began with a random dot stimulus lasting for a maximum of 1.2 s, or until the subject made a behavioural response. Subjects were instructed to respond as quickly as possible, and had a time limit of 1.5 s to do so. The message 'Oops! Too slow' was displayed if this time limit was exceeded or no direction response was made. Once the dot stimulus disappeared, the screen remained blank until the 1.2 s stimulation period elapsed and through an additional random delay (1.5 – 4 s).

Next, subjects were presented with a rating scale for 3 s, during which they reported their confidence in the previous direction decision. The confidence scale was represented intuitively by means of a white horizontal bar of linearly varying thickness, with the thick end representing high confidence. Its orientation on the horizontal axis (thin-to-thick vs. thick-to-thin) informed subjects of the response mapping, and this was equally and randomly distributed across trials to control for motor preparation effects. To make a confidence response, subjects moved an indicator (a small white triangle) along a 9-point marked line. The indicator changed colour from white to yellow when a confidence response was selected and this remained on the screen until the 3 s elapsed). A final delay (blank screen, jittered between 1.5 – 4 s) ended the trial. The timing of the inter-stimulus jitters was optimised using a genetic algorithm (*Wager and Nichols, 2003*) in order to increase estimation efficiency in the fMRI analysis. Failing to provide either a direction or a confidence response within the respective allocated time limits on a given trial rendered it invalid, and this was subsequently removed from further analyses. This resulted in a total fraction of .04 (.02 and. 02, respectively) of trials being discarded.

Subjects performed two experimental blocks of 160 trials each, corresponding to two separate fMRI runs. Each block contained two short (30 s) rest breaks, during which the MR scanner continued to run. Subjects were instructed to remain still throughout the entire duration of the experiment, including during rest breaks and in between scans. Motion coherence was held constant across trials, at the subject-specific level estimated during training. The direction of the dots was equally and randomly distributed across trials. To control for confounding effects of low-level trial-to-trial variability in stimulus properties on decision confidence, an identical set of stimuli was used in the two experimental blocks. Specifically, for each subject, the random seed, which controlled dot stimulus motion parameters in the stimulus presentation software was set to a fixed value. This manipulation allowed for subsequent control comparisons between pairs of identical stimuli.

Subjects were encouraged to explore the entire scale when making their responses and to abstain from making a confidence response on a given trial if a motor mapping error had been made (for instance, a premature or accidental button press that was inconsistent with the perceptual representation). They were instructed to make their responses as quickly and accurately as possible, and provide a response on every trial. All behavioural responses were executed using the right hand, on an MR-compatible button box.

## EEG data acquisition

EEG data was collected using an MR-compatible EEG amplifier system (Brain Products, Germany). Continuous EEG data was recorded using the Brain Vision Recorder software (Brain Products, Germany) at a sampling rate of 5000 Hz. We used 64 Ag/AgCl scalp electrodes positioned according to the 10 – 20 system, and one nasion electrode. Reference and ground electrodes were embedded in the EEG cap and were located along the midline, between electrodes Fpz and Fz, and between electrodes Pz and Oz, respectively. Each electrode had in-line 10 kOhm surface-mount resistors to ensure subject safety. Input impedance was adjusted to < 25 kOhm for all electrodes. Acquisition of the EEG data was synchronized with the MR data acquisition (Syncbox, Brain Products, Germany), and MR-scanner triggers were collected separately to enable offline removal of MR gradient artifacts from the EEG signal. Scanner trigger pulses were lengthened to 50µs using a built-in pulse stretcher, to facilitate accurate capture by the recording software. Experimental event markers (including participants' responses) were synchronized, and recorded simultaneously, with the EEG data.

## EEG data processing

Preprocessing of the EEG signals was performed using Matlab (Mathworks, Natick, MA). EEG signals recorded inside an MR scanner are contaminated with gradient artifacts and ballistocardiogram

(BCG) artifacts due to magnetic induction on the EEG leads. To correct for gradient-related artifacts, we constructed average artifact templates from sets of 80 consecutive functional volumes centred on each volume of interest, and subtracted these from the EEG signal. This process was repeated for each functional volume in our dataset. Additionally, a 12 ms median filter was applied in order to remove any residual spike artifacts. Further, we corrected for standard EEG artifacts and applied a 0.5 – 40 Hz band-pass filter in order to remove slow DC drifts and high frequency noise. All data were downsampled to 1000 Hz.

To remove eye movement artifacts, subjects performed an eye movement calibration task prior to the main experiment (with the MRI scanner turned off, to avoid gradient artifacts), during which they were instructed to blink repeatedly several times while a central fixation cross was displayed in the centre of the computer screen, and to make lateral and vertical saccades according to the position of the fixation cross. We recorded the timing of these visual cues and used principal component analysis to identify linear components associated with blinks and saccades, which were subsequently removed from the EEG data (*Parra et al., 2005*).

Next, we corrected for cardiac-related (i.e., ballistocardiogram, BCG) artifacts. As these share frequency content with the EEG, they are more challenging to remove. To minimise loss of signal power in the underlying EEG signal, we adopted a conservative approach by only removing a small number of subject-specific BCG components, using principal component analysis. We relied on the single-trial classifiers to identify discriminating components that are likely to be orthogonal to the BCG. BCG principal components were extracted from the data after the data were first low-pass filtered at 4 Hz to extract the signal within the frequency range where BCG artifacts are observed. Subject-specific principal components were then determined (average number of components across subjects: 1.8). The sensor weightings corresponding to those components were projected onto the broadband data and subtracted out. Finally, data were baseline corrected by removing the average signal during the 100 ms prestimulus interval.

## Single-trial EEG analysis

To increase statistical power of the EEG data analysis, trials were separated into three confidence groups (Low, Medium, High), on the basis of the original 9-point confidence rating scale. Specifically, we isolated High- and Low-confidence trials by pooling across each subject's three highest and three lowest ratings, respectively. To ensure robustness of our single trial EEG analysis, we imposed a minimum limit of 50 trials per confidence trial group. For those data sets where subjects had an insufficient number of trials in the extreme ends of the confidence scale, neighbouring confidence bins were included to meet this limit.

We used a single-trial multivariate discriminant analysis, combined with a sliding window approach (*Parra et al., 2005*; *Sajda et al., 2009*) to discriminate between High- and Low-confidence trials in the stimulus-locked EEG data. This method aims to estimate, for predefined time windows of interest, an optimal combination of EEG sensor linear weights (i.e., a spatial filter) which, applied to the multichannel EEG data, yields a one-dimensional projection (i.e., a 'discriminant component') that maximally discriminates between the two conditions of interest. Importantly, unlike univariate trial-average approaches for event-related potential analysis, this method spatially integrates information across the multidimensional sensor space, thus increasing signal-to-noise ratio whilst simultaneously preserving the trial-by-trial variability in the signal, which may contain task-relevant information. In our data, we identified confidence-related discriminating components, $y$(t), by applying a spatial weighting vector $w$ to our multidimensional EEG data $x$(t), as follows:

$$y(t) = w^T x(t) = \sum_{i=1}^{D} w_i x_i(t) \qquad (1)$$

where $D$ represents the number of channels, indexed by $i$, and $T$ indicates the transpose of the matrix. To estimate the optimal discriminating spatial weighting vector $w$, we used logistic regression and a reweighted least squares algorithm (*Jordan and Jacobs, 1994*). We applied this method to identify $w$ for short (60 ms) overlapping time windows centred at 10 ms-interval time points, between -100 and 1000 ms relative to the onset of the random dot stimulus (i.e., the perceptual decision phase of the trial). This procedure was repeated for each subject and time window. Applied to an individual trial, spatial filters ($w$) obtained this way produce a measurement of the discriminant

component amplitude for that trial. In separating the High and Low trial groups, the discriminator was designed to map the component amplitudes for one condition to positive values and those of the other condition to negative values. Here, we mapped the High confidence trials to positive values and the Low confidence trials to negative values, however note that this mapping is arbitrary.

To quantify the performance of the discriminator for each time window, we computed the area under a receiver operating characteristic (ROC) curve (i.e., the Az value), using a leave-one-out cross-validation procedure (*Duda et al., 2001*). Specifically, for every iteration, we used N-1 trials to estimate a spatial filter (*w*), which was then applied to the remaining trial to obtain out-of-sample discriminant component amplitudes (*y*) for High- and Low-confidence trials and compute the Az. Note that these out-of-sample *y* values were highly correlated with the *y* values resulting from the original High- vs. Low-confidence discrimination described above (subject-averaged R=.93). We determined significance thresholds for the discriminator performance using a bootstrap analysis whereby trial labels were randomised and submitted to a leave-one-out test. This randomisation procedure was repeated 500 times, producing a probability distribution for *Az*, which we used as reference to estimate the *Az* value leading to a significance level of p<0.01.

Given the linearity of our model we also computed scalp projections of the discriminating components resulting from *Equation 1* by estimating a forward model for each component:

$$a = \frac{X\,y}{y^T y} \tag{2}$$

where the EEG data (*X*) and discriminating components (*y*) are now in a matrix and vector notation, respectively, for convenience (i.e., both *X* and *y* now contain a time dimension). Equation 2 describes the electrical coupling of the discriminating component *y* that explains most of the activity in *X*. Strong coupling indicates low attenuation of the component *y* and can be visualised as the intensity of vector a.

## Single-trial power analysis

We calculated prestimulus alpha power (8 – 12 Hz) in the 400 ms epoch beginning at −500 ms relative to the onset of the random dot stimulus. To do this, we used the multitaper method (*Mitra and Pesaran, 1999*) as implemented in the FieldTrip toolbox for Matlab (http://www.ru.nl/neuroimaging/fieldtrip). Specifically, for each epoch data were tapered using discrete prolate spheroidal sequences (two tapers for each epoch; frequency smoothing of ± 4 Hz) and Fourier transformed. Resulting frequency representations were averaged across tapers and frequencies. Single-trial power estimates were then extracted from the occipitoparietal sensor with the highest overall alpha power and baseline normalised through conversion to decibel units (dB).

## Assessing the influence of neural confidence on behaviour

To test whether fluctuations in the confidence-discriminating component amplitudes, $y_{CONF}$, were predictive of the probability to repeat a choice on the immediately subsequent trial, $P_{REPEAT}$), we divided $y_{CONF}$ into 3 equal bins (Low, Medium, and High), separately for each subject, and compared the corresponding $P_{REPEAT}$ across subjects, using a one-way repeated measures ANOVA. To ensure that any observed modulation of $P_{REPEAT}$ by $y_{CONF}$ was independent of the correlation of $y_{CONF}$ with accuracy on the current trial(s), we first equalised the number of correct and error trials within each $y_{CONF}$ bin. Specifically, for each subject, we removed either exclusively correct or error trials (depending on which of the two was in excess) via random selection from 500 permutations of the trial set. We report results based on the average $y_{CONF}$ values obtained with this procedure (see Results).

## Modelling decision confidence

We modelled the perceptual decision process using a variant of the original race model of decision making (*Vickers, 1979*; *Vickers and Packer, 1982*; *De Martino et al., 2013*). Specifically, each decision was represented as a race-to-threshold between two independent accumulating signals - variables L and R - which collected evidence in favour of the left and right choices, respectively. At each time step of the accumulation (time increment = 1 ms), the two variables were updated separately

with an evidence sample s(t) extracted randomly from normal distributions with mean μ and standard deviation σ, s(t)=N(μ,σ), such that:

$$L(t+1) = L(t) + s_L(t) \tag{3}$$

$$R(t+1) = R(t) + s_R(t)$$

Here, we assumed that evidence samples for the two possible choices are drawn from distributions with identical variances but distinct means, whereby the mean of the distribution is dependent on the identity of the presented stimulus. For instance, a leftward motion stimulus would be associated with a larger distribution mean (and thus on average faster rate of evidence accumulation) in the left (stimulus-congruent) than right (stimulus-incongruent) accumulator. We defined the mean of the distribution associated with the stimulus-congruent accumulator as $\mu_{congr}$=0.1 (arbitrary units), and that of the stimulus-incongruent accumulator as $\mu_{incongr}$=$\mu_{congr}$/r, where r is a free parameter in the model. For each simulated trial, evidence accumulation for the two accumulator variables began at 0 and progressed towards a fixed decision threshold θ, with choice being determined by the first accumulator to reach this threshold. Finally, response time was defined as the time taken to reach the decision threshold plus a non-decision time (nDT) accounting for early visual encoding and motor preparation processes.

We fitted the model to each subject's response time data, using a maximum likelihood function (as in *Pisauro et al., 2017*). Namely, we combined RTs for correct and incorrect trials into a single distribution by mirroring the distribution of incorrect trials at the 0 point on the time axis, and thus transforming all error RTs into negative values. We compared resulting distributions and mean choice accuracies obtained from behavioural data vs. model simulations. The log likelihood function was estimated according to:

$$LL \sim log(KS(RT_{data}, RT_{model})) + log\left(exp\left(-\left(\frac{Accuracy_{data} - Accuracy_{model}}{0.1}\right)^2\right)\right) \tag{4}$$

KS represents the estimated probability that two independent samples (here, behavioural vs. simulated RTs) come from populations with the same distribution, as inferred with the two-sample Kolmogorov-Smirnov test (implemented in Matlab function kstest2).

For each subject, the free model parameters were iteratively adjusted to maximise the LL. This was done by performing a grid search through a fixed range of values (σ=[.6:0.1:1], θ=[55:7:97], nDT=[250:50:450], r=[1.2:0.05:1.6]), determined after an initial exploratory search which sought to identify parameter ranges that generated plausible behavioural measures (RT and accuracy) (i.e., comparable to those observed in subjects' behaviour). For each set of parameters, we simulated 500 trials and recorded mean choice accuracy, RT, and confidence (Δe).

To assess the quality of the model fits, we computed the correlation between observed vs. model-predicted behaviour (namely response time quantiles for correct and error responses, as well as mean choice accuracy), using the robust percentage bend correlation analysis (*Wilcox, 1994*).

## Exploratory mediation analysis

To examine the relationship between model-derived confidence estimates (Δe), neural confidence signals ($y_{CONF}$), and behavioural confidence reports (Ratings), we performed an exploratory mediation analysis (M3 toolbox for Matlab; *Wager, 2018* http://wagerlab.colorado.edu/tools) on these measures. A mediation analysis aims to identify whether the link between a predictor variable (here, Δe) and an outcome (Ratings) can be explained, fully or partially, by the indirect effect of a mediator variable ($y_{CONF}$). For each of the three variables of interest, we computed the mean difference between correct and error trials, and resulting values ($\Delta e_{DIFF}$, $y_{CONF\_DIFF}$, and $Ratings_{DIFF}$, respectively) were subjected to the mediation analysis. To establish significance of the mediator effect of $y_{CONF\_DIFF}$, three conditions must be met 1) $\Delta e_{DIFF}$ reliably predicts $y_{CONF\_DIFF}$, 2) $y_{CONF\_DIFF}$ reliably predicts $Ratings_{DIFF}$ when the effect of $\Delta e_{DIFF}$is accounted for, and (3) a significant indirect effect of $y_{CONF\_DIFF}$, defined as the coefficient product of effects (1) and (2), can be observed. We established coefficient significance in the three models using a 5000 sample bootstrap test (*Wager et al., 2008*).

## MRI data acquisition

Imaging was performed at the Centre for Cognitive Neuroimaging, Glasgow, using a 3-Tesla Siemens TIM Trio MRI scanner (Siemens, Erlangen, Germany) with a 12-channel head coil. Cushions were placed around the head to minimize head motion. We recorded two experimental runs of 794 whole-brain volumes each, corresponding to the two blocks of trials in the main experimental task. Functional volumes were acquired using a T2*-weighted gradient echo, echo-planar imaging sequence (32 interleaved slices, gap: 0.3 mm, voxel size: 3 × 3 × 3 mm, matrix size: 70 × 70, FOV: 210 mm, TE: 30 ms, TR: 2000 ms, flip angle: 80°). Additionally, a high-resolution anatomical volume was acquired at the end of the experimental session using a T1-weighted sequence (192 slices, gap: 0.5 mm, voxel size: 1 × 1 × 1 mm, matrix size: 256 × 256, FOV: 256 mm, TE: 2300 ms, TR: 2.96 ms, flip angle: 9°), which served as anatomical reference for the functional scans.

## fMRI preprocessing

The first 10 volumes prior to task onset were discarded from each fMRI run to ensure a steady-state MR signal. Additionally, 13 volumes were discarded from the post-task period at the end of each block. The remaining 771 volumes were used for statistical analyses. Pre-processing of the MRI data was performed using the FEAT tool of the FSL software (FMRIB Software Library, http://www.fmrib.ox.ac.uk/fsl) and included slice-timing correction, high-pass filtering (>100 s), and spatial smoothing (with a Gaussian kernel of 8 mm full width at half maximum), and head motion correction (using the MCFLIRT tool). The motion correction preprocessing step generated motion parameters which were subsequently included as regressors of no interest in the general linear model (GLM) analysis (see fMRI analysis below). Brain extraction of the structural and functional images was performed using the Brain Extraction tool (BET). Registration of EPI images to standard space (Montreal Neurological Institute, MNI) was performed using the Non-linear Image Registration Tool with a 10 mm warp resolution. The registration procedure involved transforming the EPI images into an individual's high-resolution space (with a linear, boundary-based registration algorithm [*Greve and Fischl, 2009*]) prior to transforming to standard space. Registration outcome was visually checked for each subject to ensure correct alignment.

## fMRI analysis

Whole-brain statistical analyses of functional data were conducted using a general linear model (GLM) approach, as implemented in FSL (FEAT tool):

$$Y = \beta X + \varepsilon = \beta_1 X_1 + \beta_2 X_2 + \ldots + \beta_n X_n + \varepsilon \tag{5}$$

where $Y$ represents the BOLD response time series for a given voxel, structured as a T×1 (T time samples) column vector, and $X$ represents the T×N (N regressors) design matrix, with each column representing one of the psychological regressors (see GLM analysis below for details), convolved with a canonical hemodynamic response function (double-gamma function). $\beta$ represents the parameter estimates (i.e., regressor betas) resulting from the GLM analysis in the form of a N × 1 column vector. Lastly, $\varepsilon$ is a T × 1 column vector of residual error terms. A first-level analysis was performed to analyse each subject's individual runs. These were then combined at the subject-level using a second-level analysis (fixed effects). Finally, a third-level mixed-effects model (FLAME 1) was used to combine data across all subjects.

## Simultaneous EEG-fMRI analysis

With the combined EEG-fMRI approach, we sought to identify confidence-related activation in the fMRI surpassing what could be explained by the relevant behavioural predictors alone. In particular, we looked for brain regions where BOLD responses correlated with the confidence-discriminating component derived from the EEG analysis. Our primary motivation behind this approach was the hypothesis that endogenous trial-by-trial variability in the confidence discriminating EEG component (near the time of perceptual decision, and prior to behavioural response) would be more reflective of early internal representations of confidence at the single-trial level, compared to the metacognitive reports which are provided post-decisionally and therefore likely to be subjected to additional processes. We predicted that the simultaneous EEG-fMRI approach would enable identification of latent brain states that might remain unobserved with a conventional analysis approach. To this end,

we extracted trial-by-trial amplitudes of $y(t)$ (resulting from Eq. 1) at the time window of maximum confidence discrimination, and used these to build a BOLD predictor (i.e., the $y_{CONF}$ regressor). Importantly, to avoid possible confounding effects of motor preparation/response, the time of this component was determined on a subject-specific basis, by only considering the period prior to the behavioural choice (mean peak discrimination time = 708 ms from stimulus onset, SD=162 ms). Thus, on average this was selected 287ms (SD=171 ms) prior to each subject's mean response time. To ensure our results were not affected by potential overfitting during the estimation of $y$, we conducted a control GLM analysis whereby the $y_{CONF}$ regressor was built using fully out-of-sample $y$ values resulting from our leave-one-out cross-validation procedure detailed above (*Figure 5—figure supplement 5*).

Note that the trial-by-trial variability in our EEG component amplitudes is driven mostly by cortical regions found in close proximity to the recording sensors and to a lesser extent by distant (e.g., subcortical) structures. Nonetheless, an advantage of our EEG-informed fMRI predictors is that they can also reveal relevant fMRI activations within deeper structures, provided that their BOLD activity covaries with that of the cortical sources of our EEG signal.

## GLM analysis

We designed our GLM model to account for variance in the BOLD signal at two key stages of the trial, namely the perceptual decision period (beginning at the onset of the random dot visual stimulus) and the metacognitive evaluation/rating (beginning at the onset of the rating scale display), respectively. A total of 10 regressors were included in the model. Our primary predictor of interest was the EEG-derived endogenous measure of confidence ($y_{CONF}$ regressor). We modelled this as a stick function (duration = 0.1 s) locked to the stimulus onset, with event amplitudes parametrically modulated by the trial-to-trial variability in the confidence discriminating component $y(t)$. To ensure variance explained by this regressor was unique (i.e., not explained by subjects' behavioural reports), we included a second regressor whose event amplitudes were parametrically modulated by confidence ratings, and which was otherwise identical to the $y_{CONF}$ regressor (i.e., Ratings$_{DEC}$ regressor, duration = 0.1 s, locked to stimulus onset). Importantly, $y_{CONF}$ amplitudes were only moderately correlated with behavioural confidence ratings, thus allowing us to exploit additional explanatory power inherent to this regressor. Other regressors of no interest for the perceptual decision stage included: one regressor parametrically modulated by prestimulus alpha power in the EEG signal (to control for potential attentional baseline effects), one categorical regressor (1/0) accounting for variability in response accuracy, and one unmodulated regressor (all event amplitudes set to (1) modelling stimulus-related visual responses of no interest across both valid and non-valid (missed) trials (all event durations = 0.1 s, locked to stimulus onset). To control for motor preparation/response, we also included a parametric regressor modulated by subjects' reaction time on the direction discrimination task (duration = 0.1 s, locked to the time of behavioural response). Note that including an additional unmodulated regressor locked to the time of the behavioural response did not alter our results.

Additionally, locked to the onset of the metacognitive rating period, we included one parametric regressor (duration = 0.1 s) with event amplitudes modulated by subjects' confidence ratings, one boxcar regressor with duration equivalent to subjects' active behavioural engagement in confidence rating (to minimise effects relating to motor processes), and one unmodulated regressor (duration = 0.1 s). Lastly, we included one categorical boxcar regressor (1/0) to model non-task activation (i.e., rest breaks within each run). Motion correction parameters obtained from fMRI preprocessing were entered as additional covariates of no interest.

As we included two rating-modulated regressors in our model, which were identical except for their onset times (i.e., decision and rating phases, respectively), we sought to ensure that these were not highly correlated. We computed the correlation between the convolved regressors, separately for each subject and experimental run (mean R = -.13; *Figure 5—figure supplement 3*). Additionally, we conducted two separate control GLM analyses whereby only the regressors pertaining to one trial phase (i.e., decision or rating, respectively) were included at a time. This allowed us to further validate our results, to ensure they remained unaffected by potential correlations between regressors at the two stages of the trial (*Figure 5—figure supplement 4*). Finally, we also assessed the correlations between all regressors by computing the variance inflation factors (VIF) for the regressors in our model. We found that mean VIF = 3.57 (±1.83), with multicollinearity typically being considered high if VIF > 5 – 10.

## Resampling procedure for fMRI thresholding

To estimate a significance threshold for our fMRI statistical maps whilst correcting for multiple comparisons, we performed a nonparametric permutation analysis that took into account the a priori statistics of the trial-to-trial variability in our primary regressor of interest ($y_{CONF}$), in a way that trades off cluster size and maximum voxel Z-score (*Debettencourt et al., 2011*). For each resampled iteration, we maintained the onset and duration of the regressor identical, whilst shuffling amplitude values across trials, runs and subjects. Thus, the resulting regressors for each subject were different as they were constructed from a random sequence of regressor amplitude events. This procedure was repeated 200 times. For each of the 200 resampled iterations, we performed a full 3-level analysis (run, subject, and group). Our design matrix included the same regressors of non-interest used in all our GLM analysis. This allowed us to construct the null hypothesis $H_0$, and establish a threshold on cluster size and Z-score based on the cluster outputs from the permuted parametric regressors. Specifically, we extracted cluster sizes from all activations exceeding a minimal cluster size (5 voxels) and Z-score (2.57 per voxel) for positive correlations with the permuted parametric regressors. Finally, we examined the distribution of cluster sizes (number of voxels) for the permuted data and found that the largest 5% of cluster sizes exceeded 162 voxels. We therefore used these results to derive a corrected threshold for our statistical maps, which we then applied to the clusters observed in the original data (that is, Z=2.57, minimum cluster size of 162 voxels, corrected at p=0.05).

## Psychophysiological interaction analysis

We conducted a psychophysiological (PPI) analysis to explore potential functional connectivity between the region of the VMPFC found to uniquely explain trial-to-trial variability in our electrophysiologically-derived measures of confidence, and the rest of the brain, during the perceptual decision phase of the trial. To carry out the PPI analysis, we first extracted the time-series data from the seed region. Specifically, we identified the cluster of interest at the group level (i.e., in standard space) by applying the cluster correction procedure described in the previous section. Using this as a template, we constructed subject-specific masks of the voxels exhibiting the strongest correlation with the VMPFC region of interest, and back-projected these into the functional space of each individual. Resulting masks were used to compute average time-series data, separately for each subject and functional run, which subsequently served as the physiological regressor(s) in the PPI model. To carry out the PPI analysis, we performed a new GLM analysis. This included the following regressors, locked to the time of stimulus onset: (1) an unmodulated regressor (all event amplitudes set to 1), (2) the physiological regressor (time course of the VMPFC seed), (3) the psychological regressor (a boxcar function with event amplitudes set to one and duration parametrically modulated by trial-specific decision times (i.e., interval between stimulus presentation and behavioural response on the perceptual task), and (4) the interaction regressor. Additionally, motion parameters estimated during registration (see preprocessing step) were included as regressors of no interest. The statistical output from the interaction regressor thus reveals regions of the brain where correlation with the BOLD signal in the VMPFC is stronger during the perceptual decision than the rest of the trial. Importantly, this represents variance additional to that explained by the psychological and physiological regressors alone. Correction for multiple comparisons was performed on the whole brain using the outcome of the resampling procedure as described earlier.

## Extracting BOLD response time course

To illustrate the activation time course within the VMPFC cluster identified with our EEG-informed fMRI analysis, we first extracted the average BOLD response time-series from this region, separately for each subject and functional run (as detailed in the previous section). We aligned our data to the onset of the random-dot stimulus, by approximating to the time of the nearest fMRI volume, and defined the temporal window of interest as the -4 s to 10 s interval relative to stimulus onset. We proceeded to separate trials into three bins according to the magnitude of the confidence discriminating component $y_{CONF}$ (i.e., Low, Medium, and High $y_{CONF}$), and computed the respective percent signal change as follows:

$$\%BOLD\ Change_j(t) = \frac{BOLD_j(t) - BOLD_j^{baseline}}{\overline{BOLD}} \tag{6}$$

where $j$ represents the trial index, $BOLD(t)$ represents the stimulus-locked data at time point $t$, and $BOLD^{baseline}$ is the mean baseline data, with the baseline window defined as the 4 s interval prior to stimulus onset. Finally, $\overline{BOLD}$ is the average signal across the entire functional run. Resulting signals were averaged across trials, runs, and subjects.

## Acknowledgements

This work was supported by the Economic and Social Research Council (ESRC; grant ES/L012995/1 to MGP) and the British Academy (BA; grant SG121587 to MGP).

## Additional information

### Funding

| Funder | Grant reference number | Author |
| --- | --- | --- |
| Economic and Social Research Council | ES/L012995/1 | Marios Philiastides |
| British Academy | SG121587 | Marios Philiastides |

The funders had no role in study design, data collection and interpretation, or the decision to submit the work for publication.

### Author contributions

Sabina Gherman, Conceptualization, Resources, Data curation, Software, Formal analysis, Validation, Investigation, Visualization, Methodology, Writing—original draft, Writing—review and editing; Marios G. Philiastides, Conceptualization, Resources, Software, Formal analysis, Supervision, Funding acquisition, Validation, Investigation, Visualization, Methodology, Writing—original draft, Project administration, Writing—review and editing

### Author ORCIDs

Sabina Gherman  http://orcid.org/0000-0001-9918-3692
Marios G. Philiastides  http://orcid.org/0000-0002-7683-3506

### Ethics

Human subjects: The study was approved by the College of Science and Engineering Ethics Committee at the University of Glasgow (CSE01355) and informed consent, and consent to publish, was obtained from all participants.

### Decision letter and Author response

Decision letter https://doi.org/10.7554/eLife.38293.026
Author response https://doi.org/10.7554/eLife.38293.027

## Additional files

### Supplementary files

• Supplementary file 1. Supplementary Tables - Complete list of brain activations correlating with explicit confidence reports.
DOI: https://doi.org/10.7554/eLife.38293.018

• Transparent reporting form
DOI: https://doi.org/10.7554/eLife.38293.019

## Data availability

The data and code required to reproduce the main and supplementary figures have been uploaded to Dryad. The full EEG-fMRI dataset is freely available at: https://openneuro.org/datasets/ds001512.

The following datasets were generated:

| Author(s) | Year | Dataset title | Dataset URL | Database and Identifier |
|---|---|---|---|---|
| Gherman S, Philiastides MG | 2018 | Data from: Human VMPFC encodes early signatures of confidence in perceptual decisions | https://doi.org/10.5061/dryad.3dn0045 | Dryad, 10.5061/dryad.3dn0045 |
| Gherman S, Philiastides MG | 2018 | Simultaneous EEG-fMRI - Confidence in perceptual decisions | https://openneuro.org/datasets/ds001512 | OpenNEURO, ds001512 |

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
