## [Decision Letter]

[Editors’ note: a previous version of this study was rejected after peer review, but the authors submitted for reconsideration. The first decision letter after peer review is shown below.]

Thank you for submitting your work entitled "Human VMPFC encodes early signatures of confidence in perceptual decisions" for consideration by *eLife*. Your article has been reviewed by four peer reviewers, one of whom is a member of our Board of Reviewing Editors, and the evaluation has been overseen by a Senior Editor. The reviewers have opted to remain anonymous.

Our decision has been reached after consultation between the reviewers. Based on these discussions and the individual reviews below, we regret to inform you that your work will not be considered further for publication in *eLife*.

All reviewers agreed that your study addresses an important topic, and that the simultaneous fMRI-EEG approach has high potential for doing so. However, all reviewers also raised a number of substantive concerns about the specifics of the approach, and neither of the reviewers was sufficiently convinced by the conceptual advance afforded by this study.

Specifically, the reviewers identified one central issue as limiting the conclusions that can be drawn from the results – the functional meaning of the discriminating component amplitude Y remains unclear. This is for a number of reasons: (i) there is no model that explains how Y is computed and links it to the elements of the decision process, which are better understood (sensory input, decision variable, internal noise); (ii) Y is only partially correlated with confidence ratings; (iii) the behaviour of Y is at odds with most current models of confidence (e.g. Y does not predict choice accuracy); (iv) no predictive effect of Y on future behaviour is established; and (v) by design, reaction times (and with it stimulus duration and time of response preparation) vary from trial to trial, raising concerns about trivial explanations for Y. In addition the reviewers were not convinced that the functional connectivity analysis adds anything of substance to the paper.

That said, most reviewers felt that an improved version of this paper which convincingly addresses the above concerns and the ones of the individual reviewers copied below might warrant another round of formal review. An essential prerequisite for this would be the presentation of a clear generative model for Y and a new version of the analysis and/or an improved experimental design that eliminates the reaction time confound from the decoding analysis. In such a case, we would need to evaluate the changes first to decide if further review is justified and therefore we would treat the paper as a new submission (but considering its full history).

Reviewer #1:

This is an interesting and timely study into the neural basis of perceptual decision confidence behavior. The overall approach is original and state-of-the-art. The early confidence signal uncovered in VMPFC is novel and potentially important. That said, I am troubled by a number conceptual and methodological issues.

1) One conceptual limitation is that the paper does not provide any insight into how the early confidence signal is constructed in the brain – specifically, how the confidence signal relates to sensory evidence, and the internal decision variable that the brain derives from that evidence. This issue is central to current theoretical work on perceptual decision confidence. Addressing this issue would substantially raise the significance of the paper. The functional connectivity analysis might shed some light onto this issue, but the authors should take VMPFC (rather than RLPFC, as they do now) as a seed. Then, the analysis should reveal two sets of regions: those that drive the VMPC signal (regions encoding the decision variable?) and those are driven by the VMPC signal (RLPFC, as the authors speculate?).

2) A second limitation is that the functional role of the neural confidence signals is not assessed. Many influential papers (theoretical and experimental) in this field have begun to uncover the roles of confidence in controlling behaviour and learning. While the task design is not tailored to addressing this issue, the authors could test for confidence-dependent, short-term changes in choice behavior or longer-term learning effects. Again, this would raise the significance of the findings reported.

3) The logic behind the PPI analysis needs to be unpacked – it is not clear if the result provides any conceptual insight. First, the authors seem to suggest that the VMPFC confidence signal drives RLPFC – then, why should the strength of this correlation scale with confidence? Should the correlation not be the same, regardless of whether confidence is high or low? Second, the functional consequences of this coupling result are unclear. This part of the authors' conclusions is purely speculative ("informing metacognitive evaluation and learning") – a meaningful link to behavior would help.

Reviewer #2:

In this paper, the authors describe a combined EEG-MRI study that links early predictors of confidence in a perceptual discrimination task to stimulus-locked activity in the ventromedial prefrontal cortex (vmPFC). Globally the paper is well written, with many aspects to praise in the design (notably controls for difficulty and motor confounds) and very sophisticated analyses.

To me it is not uncommon to see vmPFC activity associated with confidence in perceptual decision tasks, even if this is generally not the main message. So I think this claim is not particularly novel. However, the dataset reported here is rather unique because it allows using EEG measurements to track the neural noise that is added to perceptual evidence in the generation of confidence. This construct (called Y by the authors) can then be used as a regressor in the analysis of fMRI data.

The main issue is the circularity in the approach: the multivariate EEG decoder is trained to predict confidence ratings, and then the output of this decoder (i.e., Y) is said to represent an early predictor of confidence that is different from the rating. I am convinced that vmPFC activity indeed correlates with Y and not rating, but what is Y? That is the question. Without a precise specification of this construct, I do not think there is much information to get from the result. It remains open to uninteresting interpretations: for instance Y could represent the fact that at this time point the decision has been made or not, or the proximity of the motor response, since confidence correlates with response time.

My suggestion is to build a generative model of confidence rating, in which Y would be a factor among others. What needs to be explained is how Y is generated (possibly something like evidence plus neural noise) and then how it is transformed into choice, response time and confidence rating. If Y could be estimated independently of rating, then the circularity would be broken and the dissociation between neural representations of Y and rating would be meaningful. Computational modeling may be helpful here, perhaps a race model as that used in De Martino et al., 2013.

Another concern relates to the PPI analysis. I cannot make sense of the result that vmPFC activity reflects the interaction between confidence rating and time series in the rostrolateral PFC. If this region already signals confidence level, then the interaction regressor is something like confidence squared. Does this really tell us anything about the passage of information from vmPFC to rlPFC?

Reviewer #3:

This study investigates the neural mechanisms of perceptual decision confidence using a combination of EEG, fMRI and machine learning. Participants discriminate the drift direction of moving dot kinematograms. Motion coherence was varied on a per-subject basis to achieve around 75% performance in each participant. The perceptual task is speeded, meaning that the response is given as soon as the participant feels ready and then the stimulus presentation is terminated. There are three key analyses: The first links trials-wise reported confidence during the decision making stage directly to BOLD signals and finds the striatum, lateral OFC, ACC and other regions to be positively related to confidence, consistent with previous work. The second analysis links reported confidence during the rating stage to BOLD signals and finds striatum, medial temporal regions and motor cortex to be positively related to confidence. The third main analysis is based on a trial-wise EEG-classifier that classifies the confidence on each trial. This third component was related to BOLD signlas in ventromedial PFC.

The findings here are interesting and add to and extend the literature on confidence signals in the brain. However, I have a number of points that still need to be clarified.

1) The use of a speeded perceptual task means that the stimulus presentation duration is shorter for high confidence than for low confidence trials. I was wondering which effect this contamination effect has on the EEG classifier, and thus also in turn on the BOLD signals observed in the third main (i.e. EEG-based) analysis. Could it be that the classifier is partly picking up the effect of the longer stimulus duration?

2) I would be more upfront with approach used for defining and dissociating the different temporal stages (decision making, rating). I couldn't work this out until I reached the Materials and methods section, but it is vital to understanding the design.

3) I didn't understand the logic of the autocorrelation analysis that was used to control for attention. Please explain.

4) The summary of time series as "delay" and "peak" is too dense (Figure 2A). It would be better to show individual time courses to confirm that the data can be appropriately summarized by a delay and peak.

5) How can it be ensured that the EEG-derived measures are independent of difficulty, accuracy and attention? For this it would be necessary to assess the relationship between the EEG-measure and these behavioral properties explicitly (ideally to plot them as well).

Other comments:

In order to accord with requirements for reporting statistics the paper here should add a statement that "No explicit power analysis was conducted for determining sample size".

Reviewer #4:

In this paper, the authors investigate the correlates of confidence using single trial multivariate analyses based on EEG signals that were concurrently recorded during fMRI. Crucially, the authors derive interesting relations between BOLD-fMRI and EEG decoded signals during the decision stage (before an actual motor action is executed), thus allowing them to elegantly show in humans with high spatial and temporal specificity the neural correlates of confidence in perceptual decisions before a motor action is observed.

Overall this study offers new insights on the origins of confidence during perceptual decisions, by showing that the vmPFC also encodes an early confidence readout for this type of choices. I am a fan of the authors' methodological strategy to study human decision-making. However, for this study, my points of criticism are mainly related to the set of statistical analyses that the authors stand on to make their conclusions, which I consider should be revised. I provide some suggestions that may help to strengthen the authors' conclusions.

1) An important concern is the statistical fMRI modelling approach. The delay between stimulus response and confidence rating is extremely short if one wants to incorporate the same parametric regressors at both the decision and confidence rating stage. I appreciate that there is a jitter of 1.5-4 s, however, given this short average duration (~2.6 s, roughly corresponding to a bit more than 1 TR), I suspect that adding the confidence rating as parametric modulator at both the rating and decision stage will be highly correlated. Even if FSL allows to run such model, highly correlated regressors (after convolution with the HRF) can have dramatic effects on the beta weights due to variance inflation (see for instance, Mumford et al., 2015).

On a related issue, the authors write: "we also included a parametric regressor modulated by subjects' reaction time on the direction discrimination task (duration = 0.1 s, locked to the time of behavioural response)". First, did the authors also include the main effect regressor? This is not reported. If it wasn't included, then the model is wrongly specified. You cannot include a parametric regressor without including the main effect regressor. In any case, if this main effect regressor is indeed included, then once more, I suspect that this main effect regressor will be highly correlated with the main effect regressor that is included on trial onset. The mean response times are less than 1 s. If you convolve two stick functions that are less than 1 s apart, the resulting convolved regressors will be highly correlated.

I would like to see (and I think this should be formally included as supplementary information in the manuscript) for the current design, an example of the design and design_cov figures produced by FLS for two or three subjects (or in general that the authors report the correlation not only between original the parametric regressors, but also the correlations between all the regressors including both main effects and parametric regressors after convolution with the HRF).

Regarding the first point, I understand that the authors wanted to split explanatory variance between the decision and rating stages, but what I recommend and find more appropriate given my above-mentioned concern (if the authors insist in investigating the separation of confidence between the decision and the rating stages) is to run to separate GLMs, one with the parametric regressor on the decision stage, and one with the parametric regressor on the confidence rating stage and investigate whether the main conclusions of the authors still hold. Then, in a second level analysis the authors could investigate at which time point (decision or rating) there is a stronger relationship between the confidence ratings and BOLD responses.

2) In the results presented in Figure 2B, if I understood correctly, the authors report the y(t) out-of-sample values of the "unseen" data for the middle confidence level. I am not convinced that the results reported by the authors are strong evidence for their decoder's ability to generate sensible out-of-sample y(t) values as this could simply reflect regression to the mean (if the authors use the middle confidence level for reading out y(t) by using a dichotomous decoder). I do not think that this result is especially revealing nor necessary to conduct the subsequent fMRI analysis (see my next point).

3) Regarding the use of the decoded value y(t) as parametric regressor for the fMRI analysis, if I understood correctly, the authors use out of sample values of y(t) only for the middle confidence level (see my concern in the point above), whereas for the low and high confidence levels this was not the case. I think that a more appropriate analysis would be to obtain values y(t) fully out of sample. The authors can split the data in n-folds (for instance 10) and use n-1 folds to train the data and obtain the y_CONF_ regressor using the remaining fold for decoding, and then use these y_CONF_ values as regressors of the fMRI data (see for instance for a similar n-fold cross-validation approach: van Bergen et al., 2015 Nature Neuroscience). Given the nature of the decoder used by the authors (dichotomous predictions), it should be enough that the authors split the data in high and low confidence levels to train the decoder (and therefore it is not necessary to use three levels or more).

4) Subsection “EEG-derived measure of confidence”, last paragraph: Maybe I missed the point, but for me it is not entirely clear why it is expected that the discriminant component amplitudes are not different for correct and incorrect answers. One of the well established statistical signatures of confidence is that confidence is markedly different for correct and incorrect responses (e.g. see Sanders, Hangya and Kepecet al., 2016; Urai, Braun and Donner, et al.2017). Why didn't the authors expect a separation (see my next point for a related concern), and if not, what the discriminant component amplitude really reflects? More discussion on this point in general would be great.

5) Subsection “Stimuli and task”, last paragraph: On a related point (and perhaps an important caveat of this study), I do not understand why the authors excluded or explicitly asked the participants to "abstain from making a confidence response on a given trial if they became aware of having made an incorrect response". Again, one of the well established signatures of confidence is that confidence is markedly different for correct and incorrect responses. How the results would have been affected without such explicit instruction to the participants? In my opinion, this confidence information should not be excluded or spuriously biased via instructions to the participants. Therefore, I am afraid that what the authors are capturing with their actual confidence ratings (and therefore the decoded values y(t)) is a biased response that is not formally confidence per se. I urge the authors to report this potential caveat and make a clear case (from the beginning) of why this strategy was adopted in the first place.

[Editors’ note: what now follows is the decision letter after the authors submitted for further consideration.]

Thank you for submitting your article "Human VMPFC encodes early signatures of confidence in perceptual decisions" for consideration by *eLife*. Your article has been reviewed by three peer reviewers, including Tobias H Donner as the Reviewing Editor and Reviewer #1, and the evaluation has been overseen by Sabine Kastner as the Senior Editor.

The reviewers have discussed the reviews with one another and the Reviewing Editor has drafted this decision to help you prepare a revised submission.

Summary:

The authors have made substantial new analyses in their revision, and I acknowledge the paper has improved on the methodological level. Again we are impressed by the technical achievement that clearly establishes a link between the EEG-derived construct and fMRI activity. However, reviewers are still concerned with the interpretation of this construct at a functional/cognitive level. The authors fitted response time distributions for correct and error trials with a race model. The reviewers appreciate the effort made to address these concerns, but are not sufficiently convinced by the insight this provides into the underlying mechanisms.

Major comments:

1) One issue is that the model makes the same predictions for every trial, as task difficulty (motion coherence) is held constant. So the authors are bound to between-subject correlations, which they found indeed between modeled sensory evidence at the bound (Δe) and EEG-derived predictor of confidence rating (Y). My understanding of the current interpretation is that Y represents evidence plus some neural noise, and then that confidence rating (R) is Y plus something else (perhaps noise again) which could be loosely defined as 'meta-cognitive reappraisal'. This is not very informative, and not even directly tested.

A more straightforward test would be a mediation analysis, assessing whether Y could indeed mediate the link from Δe to R. The alternative hypothesis to be discarded is that Δe is actually closer to R, which would take us back to the fundamental question of how Y can be specified in cognitive terms.

Yet a more informative use of the race model would be to fit trial-by-trial variations in R. This means allowing free parameters to vary across trials, and to test their potential relationship with Y. It could be for instance that Y fluctuations arise from variations in the starting point, or, within a Bayesian framework, in the prior on motion direction. Having said this, my agenda is not to bury the paper under requests for additional work. A clarified relationship between Δe, Y and R may be a reasonable limit to what can be inferred from the dataset.

2) A related issue comes with the new analysis provided to substantiate a role for confidence in behavioral control. The authors found that higher confidence predicts repetition of the same choice in the next trial, if motion direction is the same as in the current trial. This is not in line with the computational model in its present form. It could mean that confidence influences the prior on motion direction, but this would obviously not be adaptive. There is therefore a need to reconcile this finding with the generative model of choice and confidence.

3) Y is not different for correct/incorrect responses: The authors should add a figure plotting Y split for correct and incorrect responses (perhaps next to the new panel 2C) indicating the quantitative difference. Also, please add an explicit explanation (in the Results/Discussion section) about this result. This is essential, as I believe it reveals quite a lot about what type of information Y carries, namely, it is not the classical statistical signature of confidence (see Sanders and Urai work) but something different in line to the arguments that the authors give at the end of the response to this point, i.e. not the probability that the choice is correct, but something else. The authors try to describe this "dissociation" to some extent in other parts of the text, but should be more explicit about this point.

4) Please add the results of the neural correlates of Y at the rating stage (can be in the supplement), and briefly comment about this in the Results section. Reviewers understood that the authors did not want that to focus on this stage, but it is quite a nice result that information about Y transitions from vmPFC at the decision stage to motor related areas at the confidence rating stage.

---

## [Author Response]

[Editors’ note: the author responses to the first round of peer review follow.]

Reviewer #1:This is an interesting and timely study into the neural basis of perceptual decision confidence behavior. The overall approach is original and state-of-the-art. The early confidence signal uncovered in VMPFC is novel and potentially important. That said, I am troubled by a number conceptual and methodological issues.1) One conceptual limitation is that the paper does not provide any insight into how the early confidence signal is constructed in the brain – specifically, how the confidence signal relates to sensory evidence, and the internal decision variable that the brain derives from that evidence. This issue is central to current theoretical work on perceptual decision confidence. Addressing this issue would substantially raise the significance of the paper.

We agree that a better understanding of the mechanisms underlying these early confidence signals is essential for linking theoretical and empirical work on decision confidence. In the revised manuscript, we have employed a computational modelling approach to address this question, and provide a prospective link between observed neural confidence signals and the perceptual decision process.

Specifically, we fitted our behavioural data with a variant of the race model of decision making (Vickers, 1979; Vickers and Packer, 1982; De Martino et al., 2013) (Materials and methods, subsection “Modelling decision confidence”, first paragraph), which describes the decision process as a stochastic accumulation of perceptual evidence over time by two independent signals representing the possible choices (with confidence represented as the difference in the evidence accumulated towards the two choices at the termination of the decision process – Δe). Overall the model fitted our behavioural data well, and importantly, we found that our neural measures of confidence (EEG-derived discriminant component – Y) were able to capture patterns in the model estimates of confidence (Δe) across participants (Results, subsection “Dynamic model of decision making”, last paragraph). In particular, for each subject, we computed the mean confidence difference between correct and error trials, as reflected by the neural signals (Y) and the model estimates (Δe) and tested the extent to which these quantities were correlated across subjects. This relative measure, which captured the relationship between confidence and choice accuracy, also ensured that any potential between-subject differences in the overall magnitude of the discriminant component (e.g., due to across-subject variability in overall EEG power) were subtracted out. Indeed, we found a significant positive correlation, such that subjects who showed stronger Y difference between correct and error trials also showed higher correct vs. error difference in Δe (R=.48, p=.019, robust correlation coefficient obtained using the percentage bend correlation analysis (Wilcox, 1994); see Figure 3D), suggesting that neural confidence could arise from a race-like process similar to that implemented by the current model. In other words, it is possible that, in line with the “balance of evidence hypothesis” (Vickers et al., 1979) and the idea that confidence emerges from the process of decision formation itself (Kiani and Shadlen, 2009; Gherman and Philiastides, 2015), the observed early EEG-derived measures of confidence (Y) may reflect the difference in the evidence accumulated towards the two choices at the time of decision. Alternatively, Y could also represent a (potentially noisy) readout of this difference (e.g., by a distinct system than the one supporting the perceptual choice itself).

How these early signatures of confidence contribute to post-decisional metacognitive signals and eventual confidence reports remains an open question that might be more adequately addressed with specifically tailored experimental designs (for example, by explicitly interrogating the transfer of information between networks associated with decisional and post-decisional confidence; e.g., (Fleming et al., 2018). We now discuss this issue in the seventh paragraph of the Discussion.

The functional connectivity analysis might shed some light onto this issue, but the authors should take VMPFC (rather than RLPFC, as they do now) as a seed. Then, the analysis should reveal two sets of regions: those that drive the VMPC signal (regions encoding the decision variable?) and those are driven by the VMPC signal (RLPFC, as the authors speculate?).

We thank the reviewer for this suggestion. We now report results from a separate PPI analysis where we examined whole-brain functional connectivity using the VMPFC as seed (Materials and methods, subsection “Psychophysiological interaction analysis”). In particular, we sought to identify regions that might increase their connectivity (i.e., show stronger signal correlation) with the VMPFC seed during the decision phase of the trial (defined as the time interval between stimulus presentation and subjects’ behavioural expression of choice), relative to baseline. Based on existing literature showing negative BOLD correlations with confidence ratings in regions recruited post-decisionally (e.g., during explicit metacognitive report), such as the anterior prefrontal cortex (Fleming et al., 2012; Hilgenstock et al., 2014; Morales et al., 2018), we expected that increased functional connectivity of such regions with the VMPFC would be reflected in stronger negative correlation in our PPI.

With respect to potential functional connectivity with regions involved in perceptual decision making, we hypothesised that fMRI activity in regions encoding the decision variable would correlate negatively with confidence, in line with the idea that easier (and thus more confident) decisions are characterised by faster evidence accumulation to threshold (Shadlen and Newsome, 2001) and weaker fMRI signal in reaction time tasks (Ho et al., 2009; Kayser et al., 2010; Liu and Pleskac, 2011; Filimon et al., 2013; Pisauro et al., 2017). Accordingly, we expected that if such regions increased their functional connectivity with the VMPFC during the decision, this would also manifest as stronger negative correlation in the PPI analysis.

We found increased negative correlations with the VMPFC signal in the orbitofrontal cortex (OFC), left anterior PFC (aPFC), and right dorsolateral PFC (dlPFC), shown in updated Figure 6. Regions of the aPFC and dlPFC, in particular, have been previously been linked to perceptual decision making (Noppeney et al., 2010; Liu and Pleskac, 2011; Filimon et al., 2013), as well as post-decisional confidence-related processes (Fleming et al., 2012; Hilgenstock et al., 2014; Morales et al., 2018) and metacognition (Fleming et al., 2010; Rounis et al., 2010; McCurdy et al., 2013). We now report these results in the subsection “Psychophysiological interaction (PPI) analysis”, and discuss their potential involvement with decisional confidence in the seventh paragraph of the Discussion.

2) A second limitation is that the functional role of the neural confidence signals is not assessed. Many influential papers (theoretical and experimental) in this field have begun to uncover the roles of confidence in controlling behaviour and learning. While the task design is not tailored to addressing this issue, the authors could test for confidence-dependent, short-term changes in choice behavior or longer-term learning effects. Again, this would raise the significance of the findings reported.

We thank the reviewer for this comment, and as suggested, we have conducted a series of additional analyses to address this matter. We discuss these below.

We first tested for potential influences of confidence signals on short-term decision-related behaviour. Two recent studies have shown that confidence, as captured by behavioural (Braun et al., 2018) or physiological (Urai et al., 2017) correlates, can play a role in modulating history-dependent choice biases. We thus asked whether the neural confidence signals derived from our EEG discrimination analysis might show a similar influence on subjects’ choices.

Specifically, we tested whether trial-to-trial fluctuations in the confidence discriminant component amplitudes (Y_CONF_) were predictive of the probability to repeat a choice on the immediately subsequent trial (P_REPEAT_). To this end, we divided Y_CONF_ into 3 equal bins (Low, Medium, and High) and compared the associated P_REPEAT_ across subjects. While we found no overall significant links between Y_CONF_ and subsequent choice behaviour when considering the entire data set, we did observe a positive relationship between Y_CONF_ and P_REPEAT_*if* stimulus motion on the immediately subsequent trial was in the same direction as in the current trial (one-way repeated measures ANOVA, F(2,46)=5.89, p=.005, with post-hoc tests showing a significant difference in the probability to repeat a choice after Low vs. High Y_CONF_ trials, p=.015, Bonferroni corrected; Figure 2F). Note that for this analysis, we first equalised the number of correct and error trials within each Y_CONF_ bin. This ensured that any observed modulation of P_REPEAT_ by Y_CONF_ was independent of the correlation of Y with accuracy on the current trial(s). Specifically, for each subject, we removed either exclusively correct or error trials (depending on which of the two was in excess) via random selection from 500 permutations of the trial set. Results reported here are based on the average Y values obtained with this procedure.

We found that stronger confidence signals were associated with an increased tendency to repeat the previous choice. However, there was no modulatory effect of Y_CONF_ on choice repetition/alternation behaviour when the direction of motion on the current trial differed from that of the previous trial.

Thus, choices were only affected by previous confidence when no change in motion direction had occurred from one trial to the next. Interestingly, this suggests that subjects might be able to detect consistency with the previous stimulus without necessarily having full conscious access to the motion direction of the current stimulus, which in turn impacts the modulatory effect of previous choice confidence on subjects’ tendency to repeat their choice. We now report this analysis in the subsection “Confidence-dependent influences on behaviour”.

In a separate set of analyses, we asked whether confidence in a choice might influence subjects’ decision times on subsequent trials. Error monitoring research indicates that individuals tend to respond more slowly after having committed an error (Dutilh et al., 2012), an effect known as “post-error slowing” and thought to indicate an increase in caution. We tested whether low confidence (as captured by both subjective ratings and EEG-derived neural signatures of confidence) might have a similar impact on response time slowing, however we found no evidence for such an effect. One reason could be that response time slowing occurs when one is more confident about having made an error than a correct response (i.e., when estimated probability of being correct is below chance), whereas our behavioural results suggest this was likely a rare occurrence (the lowest confidence ratings were on average associated with chance or above-chance performance on the perceptual choice). We note that these results may differ under stronger speed emphasis (the time response limit in current paradigm was 1.35 s, with the mean response time across subjects being 994 ± 35 ms). Should the reviewer deem it necessary, we would be happy to report these analyses in the revised manuscript.

With respect to the potential role of confidence on learning, we wish to emphasise that (as the reviewer has also pointed out) the design of our behavioural paradigm was not optimised for addressing this question. In fact, we designed our experiment to specifically minimise perceptual learning effects, to avoid potential confounds with confidence (e.g., lower confidence and higher confidence trials clustering towards the beginning and towards the end of the experiment respectively, which in turn could have resulted in trivial EEG discrimination performance of low-vs.-high confidence trials due to overall signal changes in the course of the experiment – e.g., impedance changes, signal adaptation, etc.). In particular, subjects underwent task training prior to participating in the simultaneous EEG/fMRI experiment, which partly served to allow subjects’ performance to reach a plateau. Though perceptual learning can also be assessed using paradigms that maintain performance constant through online adjustments of the stimulus difficulty, we opted against such an approach to avoid potential confounding effects of stimulus difficulty on confidence.

Thus, as expected, only small to no improvements can be observed in subjects’ behavioural performance over the course of the task (e.g., mean difference in the proportion of correct responses between the first and second halves of the task =.03). Similarly, we found no significant increase in confidence ratings or neural confidence signals (Y_CONF_) across trials.

Recent work suggests that confidence may act as an implicit (expected) reward signal and be used in the computation of prediction errors (i.e., the difference between expected and currently experienced reward) (Lak et al., 2017; Colizoli et al., 2018), thus guiding a reinforcement-based learning mechanism. Relatedly, confidence prediction error (the difference between expected and experienced confidence) has been hypothesised to act as a teaching signal and guide learning in the absence of feedback (Guggenmos et al., 2016). In the brain, this could potentially be implemented through a mechanism of strengthening or weakening information processing pathways that result in high and low confidence, respectively (Guggenmos and Sterzer, 2017). Though testing this hypothesis extends beyond the scope of the current study (see previous paragraph on purposely “clumping” learning effects), we might expect that fluctuations in expected vs. actual confidence signals as derived from the EEG data have a similar influence on perceptual learning. We now discuss this point in the Discussion (eighth paragraph).

3) The logic behind the PPI analysis needs to be unpacked – it is not clear if the result provides any conceptual insight. First, the authors seem to suggest that the VMPFC confidence signal drives RLPFC – then, why should the strength of this correlation scale with confidence? Should the correlation not be the same, regardless of whether confidence is high or low? Second, the functional consequences of this coupling result are unclear. This part of the authors' conclusions is purely speculative ("informing metacognitive evaluation and learning") – a meaningful link to behavior would help.

This is a valid point and we have aimed to rectify this issue in the revised manuscript by conducting a separate PPI analysis (see Materials and methods subsection “Psychophysiological interaction analysis”, and our earlier response) where we removed the parametric modulation by confidence when searching for functional connectivity with our seed region. Specifically, we now use the VMPFC region (which showed modulation by neural confidence in our original GLM analysis) as a PPI seed, and searched instead for regions across the brain where connectivity (i.e., BOLD signal correlation) increased during the decision phase of the trial, which we defined as the interval between stimulus presentation and behavioural choice.

As we note above, we found increased negative correlations with the VMPFC signal in the OFC, left aPFC, and right dlPFC. Regions of the aPFC and dlPFC have been linked to perceptual decision making (Ho et al., 2009; Noppeney et al., 2010; Liu and Pleskac, 2011; Filimon et al., 2013; Pisauro et al., 2017), as well as post-decisional confidence-related processes (Fleming et al., 2012; Hilgenstock et al., 2014; Morales et al., 2018) and metacognition (Fleming et al., 2010; Rounis et al., 2010; McCurdy et al., 2013).

Reviewer #2:[…] The main issue is the circularity in the approach: the multivariate EEG decoder is trained to predict confidence ratings, and then the output of this decoder (i.e., Y) is said to represent an early predictor of confidence that is different from the rating.

The reviewer is correct in that the EEG-derived measures of confidence rely on a classification analysis between Low- and High-confidence trials as defined by subjects’ behavioural ratings. However, we wish to clarify that the EEG classifier is not trained to predict confidence ratings per se. Rather, we make use of subjects’ ratings only for the purpose of extracting the Low- and High-confidence trial groups and training the classifier, but subsequently rely only on the single-trial graded measures of “neural” confidence (Y) to make any subsequent inferences.

The underlying assumption is that, while ratings per se may not be entirely faithful representations of early confidence signals, they may carry sufficient explanatory power to reliably estimate a set of spatial weights representing the topographical contributions to confidence signals *at the time of decision.* Importantly, the classification output Y, obtained by subjecting the original multichannel through these neural generators, will depart from the behavioural measures of confidence in that it will contain trial-to-trial information about neural signals generated by these sources, thus potentially offering additional insight into the internal processes that underlie confidence at these early stages of the decision. We now clarify these points in the Results section (subsection “EEG-derived measure of confidence”, second paragraph).

I am convinced that vmPFC activity indeed correlates with Y and not rating, but what is Y? That is the question. Without a precise specification of this construct, I do not think there is much information to get from the result. It remains open to uninteresting interpretations: for instance Y could represent the fact that at this time point the decision has been made or not, or the proximity of the motor response, since confidence correlates with response time.My suggestion is to build a generative model of confidence rating, in which Y would be a factor among others. What needs to be explained is how Y is generated (possibly something like evidence plus neural noise) and then how it is transformed into choice, response time and confidence rating. If Y could be estimated independently of rating, then the circularity would be broken and the dissociation between neural representations of Y and rating would be meaningful. Computational modeling may be helpful here, perhaps a race model as that used in De Martino et al., 2013.

We thank the reviewer for this thoughtful comment, which led us to the addition of a computational modelling component and inclusion of additional control analyses. We recognise the importance of providing a more concrete interpretation of the neural mechanisms that generate the observed confidence signals and have addressed this question more thoroughly as detailed below.

Firstly, with regards to the possibility that our EEG-derived measures of confidence (Y) might merely represent the termination of a decision or proximity of the motor response, we note that Y was only weakly correlated with subject’s response times (subject-averaged R=-.15; we now report this in Results, p. 11). In addition, Ys were extracted on average at least 100ms (mean 271 ± 162 ms) prior to subjects’ mean response times to minimise potential confounds with activity related to motor execution (due to increase in corticospinal excitability during this period (Chen et al., 1998)).

To control for potentially confounding effects of motor response in our fMRI analysis, we included a regressor which is parametrically modulated by subjects’ response times on the perceptual task. We reasoned that this regressor would absorb any variance related to motor planning and execution processes.

Finally, in our previous EEG work on decision confidence (Gherman and Philiastides, 2015) we used a delayed-response behavioural paradigm in which subjects were unaware of the mapping between choice and response effector whilst they made their perceptual decision. In that study we could still observe the same neural signature of confidence (i.e., consistent in terms of both timing and scalp topography). On the basis of the points above, we argue that neural measures Y are unlikely to be merely explained by motor-related processes.

Most importantly, in order to tackle the question of how confidence signals might emerge, we have used a computational modelling approach, as the reviewer suggested above. Specifically, we fitted our behavioural data with a variant of the race model of decision making (Vickers, 1979; Vickers and Packer, 1982; De Martino et al., 2013) (Materials and methods, subsection “Modelling decision confidence”, first paragraph).

This class of models describes the decision process as a stochastic accumulation of perceptual evidence over time by independent signals representing the possible choices. The decision terminates when one of the accumulators reaches a fixed threshold, with choice being determined by the winning accumulator. Importantly, confidence for binary choices can be estimated in these models as the absolute distance (Δe) between the states of the two accumulators at the time of decision (i.e., “balance of evidence” hypothesis).

In our model, the state of the accumulator is represented by two variables, L and R, which collect evidence in favour of the left and right choices, respectively (Figure 3A). At each time step of the accumulation, the two variables are updated separately with an evidence sample s(t) extracted randomly from a normal distributions with mean μ and standard deviation σ, s(t)=N(μ,σ), such that:

L(t+1) = L(t) + s_L_(t)

R(t+1) = R(t) + s_R_(t)

Here, we assumed that evidence samples for the two possible choices are drawn from distributions with identical variances but distinct means, whereby the mean of the distribution is dependent on the identity of the presented stimulus. For instance, a leftward motion stimulus will be associated with a larger distribution mean (and thus on average faster rate of evidence accumulation) in the left (stimulus-congruent) than right (stimulus-incongruent) accumulator. We defined the mean of the distribution associated with the stimulus-congruent accumulator as μ_congr_=0.1 (arbitrary units), and that of the stimulus-incongruent accumulator as μ_incongr_=μ_congr_/r, where r is a free parameter in the model. For each trial, evidence accumulation for the two accumulator variables begins at 0 and progresses towards a fixed decision threshold θ. Finally, response time is defined as the time taken to reach the decision threshold plus a non-decision time (nDT) which accounts for early visual encoding and motor preparation processes.

We illustrate model fits in Figure 3C (with individual subject fits shown in Figure 3—figure supplements 1 and 2). Response time distributions for correct and error trials are summarised separately using 5 quantile estimates of the associated cumulative distribution functions (Forstmann et al., 2008). Overall, we found that this model provided a good fit to the behavioural data (Accuracy: R=.76, p<.001, Figure 3B; RT: subject-averaged R=.965, all p<=.0016)

Crucially, we proceeded to inspect the relationship between our neural measures of confidence (EEG-derived discriminant component Y) and the confidence estimates predicted by the decision model (Δe) at the subject group level. Specifically, for each subject, we extracted the mean difference in confidence (as reflected by Y and Δe, respectively) between correct and error trials. We then tested the extent to which these quantities correlated across subjects. This relative measure, which captured the relationship between confidence and choice accuracy, also ensured that any potential between-subject differences in the overall magnitude of the discriminant component Y (e.g., due to across-subject variability in overall EEG power) were subtracted out. We found a significant positive correlation (i.e., subjects who showed stronger difference in Y between correct and error trials also showed a higher difference in Δe, R=.48, p=.019, see Figure 3D), opening the possibility that neural confidence signals might arise directly from a process similar to the race-like dynamic implemented by the current model.

Two possible interpretations of Y may be proposed based on our modelling results. Namely, in following the “balance of evidence hypothesis” (Vickers et al., 1979), the observed early EEG-derived measures of confidence may reflect the difference in the evidence accumulated towards the two choices at the time of decision, consistent with the idea that confidence emerges from the process of decision formation itself (Kiani and Shadlen, 2009; Gherman and Philiastides, 2015). An alternative interpretation is that Y represents a (potentially noisy) readout of this difference (i.e., by a distinct system than the one supporting the perceptual choice itself).

How these early signatures of confidence contribute to post-decisional metacognitive signals and eventual confidence reports remains an open question that might be more adequately addressed with specifically tailored experimental designs (for example, by explicitly interrogating the transfer of information between networks associated with decisional and post-decisional confidence). We discuss these interpretations in the seventh paragraph of the Discussion.

Another concern relates to the PPI analysis. I cannot make sense of the result that vmPFC activity reflects the interaction between confidence rating and time series in the rostrolateral PFC. If this region already signals confidence level, then the interaction regressor is something like confidence squared. Does this really tell us anything about the passage of information from vmPFC to rlPFC?

We thank the reviewer for their comment. To address this point, we have conducted a new PPI analysis (see Materials and methods subsection “Psychophysiological interaction analysis”) where we use the VMPFC region as a seed (as kindly suggested by two of the reviewers), and have removed the parametric modulation by confidence from the psychological regressor. Specifically, we searched for potential regions across the brain which may increase their connectivity with confidence-encoding VMPFC during the decision phase of the trial (defined as the interval between stimulus presentation and behavioural choice) relative to baseline, such as those involved in the formation of the decision and/or metacognition.

Based on existing literature showing negative BOLD correlations with confidence ratings in regions recruited post-decisionally (e.g., during explicit metacognitive report), such as the anterior prefrontal cortex (Fleming et al., 2012; Hilgenstock et al., 2014; Morales et al., 2018), we expected that increased functional connectivity of such regions with the VMPFC would be reflected in stronger negative correlation in our PPI.

Similarly, we hypothesised that fMRI activity in regions encoding the perceptual decision would also correlate negatively with confidence / VMPFC activation, in line with the idea that easier (and thus more confident) decisions are characterised by faster evidence accumulation to threshold (Shadlen and Newsome, 2001) and weaker fMRI signal in reaction time tasks (Ho et al., 2009; Kayser et al., 2010; Liu and Pleskac, 2011; Filimon et al., 2013; Pisauro et al., 2017). Accordingly, we expected that if such regions increased their functional connectivity with the VMPFC during the decision, this would manifest as stronger negative correlation in the PPI analysis.

We found increased negative correlations with the VMPFC signal in the orbitofrontal cortex (OFC), left anterior PFC (aPFC), and right dorsolateral PFC (dlPFC), shown in updated Figure 6. Regions in the aPFC and dlPFC have been linked to perceptual decision making (Noppeney et al., 2010; Liu and Pleskac, 2011; Filimon et al., 2013), as well as post-decisional confidence-related processes (Fleming et al., 2012; Hilgenstock et al., 2014; Morales et al., 2018) and metacognition (Fleming et al., 2010; Rounis et al., 2010; McCurdy et al., 2013)

Reviewer #3:[…] The findings here are interesting and add to and extend the literature on confidence signals in the brain. However, I have a number of points that still need to be clarified.1) The use of a speeded perceptual task means that the stimulus presentation duration is shorter for high confidence than for low confidence trials. I was wondering which effect this contamination effect has on the EEG classifier, and thus also in turn on the BOLD signals observed in the third main (i.e. EEG-based) analysis. Could it be that the classifier is partly picking up the effect of the longer stimulus duration?

We thank the reviewer for pointing out this potential confound. We now performed additional analyses to address it more directly.

We first investigated the correlation between our EEG-derived measures of confidence (Y) and the duration of the visual stimulus (which, as the reviewer points out, is equal to subjects’ response time). We reasoned that if stimulus presentation time had an influence on the classification analysis and consequently on the estimation of Y, we might expect these two measures to be highly correlated. However, we found only a weak correlation between Y and stimulus duration (subject-averaged R=-.15), suggesting that classification results could not have been solely driven by this factor. We report this additional analysis in the revised manuscript (Results, subsection “EEG-derived measure of confidence”, seventh paragraph). Please also note that in a previous study from our lab (Gherman and Philiastides, 2015) in which we used EEG alone to temporally characterise decision confidence, the duration of stimulus presentation was fixed at.1 s (this was followed by a forced delay of 2-2.5s prior to response, during which subjects were not aware of the mapping between stimulus and motor response). Importantly, we observed a similar temporal profile and scalp topography in a Low-vs.-High confidence discrimination.

With regards to the impact on the fMRI data, we first wish to clarify that the three sets of fMRI results we report in relation to the neural correlates of confidence were obtained using a single GLM model, which included regressors for each of our variables of interest (namely, confidence reports at the time of decision and rating, respectively, and EEG-derived confidence measures), as well as additional nuisance regressors. Crucially, we included a regressor parametrically modulated by stimulus duration (i.e., response time) which served to regress out potential variance shared with the EEG-derived regressor (note that parameter estimates obtained with standard GLM analysis in FSL reflect variability that is unique to each regressor, thus ignoring common variability, Mumford et al., 2015). We have amended the text to make the above points explicit (Results subsection “fMRI correlates of confidence”).

2) I would be more upfront with approach used for defining and dissociating the different temporal stages (decision making, rating). I couldn't work this out until I reached the Materials and methods section, but it is vital to understanding the design.

We have added explicit definitions of the “decision” and “rating” phases of the trial in the Results (subsection “fMRI correlates of behavioural confidence reports”).

3) I didn't understand the logic of the autocorrelation analysis that was used to control for attention. Please explain.

The aim of the autocorrelation analysis was to test for potential sustained fluctuations of attention that span multiple trials and might therefore be reflected in serial dependencies of either behaviour (e.g., choice) (De Martino et al., 2013) or neural signals. In particular, we were interested in ruling out the possibility that such attentional fluctuations might be the driving factor behind the variability in our EEG-derived confidence measures (Y). We expected that if that were the case, Y values on a given trial would be reliably predicted by those observed in the immediately preceding trials. However the regression model we used to test for this possibility explained only a small fraction of the variance in our Ys (subject-averaged R^2^ =.03; Results subsection “EEG-derived measure of confidence”, eighth paragraph). We have amended the text to clarify the autocorrelation analyses (Results, subsection “Behaviour”, last paragraph and subsection “EEG-derived measure of confidence”, eighth paragraph).

4) The summary of time series as "delay" and "peak" is too dense (Figure 2A). It would be better to show individual time courses to confirm that the data can be appropriately summarized by a delay and peak.

We have updated our figure to contain the time course of the confidence discrimination performance (Az) for individual subjects (Figure 2A). Note that on average we only considered peaks occurring at least 250ms after stimulus onset (to avoid early visual processes) and 100ms (mean 271 ± 162 ms) prior to subjects’ average response times to minimise potential confounds with activity related to motor execution (due to a sudden increase in corticospinal excitability in this period (Chen et al., 1998)) (Materials and methods).

5) How can it be ensured that the EEG-derived measures are independent of difficulty, accuracy and attention? For this it would be necessary to assess the relationship between the EEG-measure and these behavioral properties explicitly (ideally to plot them as well).

To control for confounding effects of difficulty on the neural measures of confidence, we maintained the motion coherence (i.e., difficulty) of the visual stimuli constant across the entire experiment, and for each subject. In addition, each stimulus in the first half of the experiment was presented again (in an identical form) in the second half of the experiment. This enabled us to compare both behavioural and neural responses to identical stimuli and further assess whether subjects might have been sensitive to subtle differences in low-level physical properties of the stimulus that go beyond motion coherence (e.g., the motion dynamics of individual dots). Importantly, we found no correlation between the EEG-derived confidence measures (i.e., confidence-discriminating component amplitudes, Y) associated with the two sets of identical stimuli (subject-averaged R=.02) (Results subsection “EEG-derived measure of confidence”, last paragraph). We believe these observations represent strong evidence that the EEG-derived measures of confidence are independent of objective difficulty.

We have also performed a control analysis to verify whether our EEG-derived confidence measures are independent of accuracy. Namely, Figure 2D illustrates that these neural signatures continue to show significant modulation by (reported) confidence when accuracy is constant (i.e., when only correct trials are considered).

Finally, we tested for potential attentional effects on EEG-derived confidence measures as follows. Firstly, we investigated the influence of occipitoparietal prestimulus α, a neural signal thought to correlate with attention and predict visual discrimination (Thut et al., 2006; van Dijk et al., 2008), on the EEG-derived confidence measures. We found that Y measures associated with High vs. Low prestimulus alpha power did not differ significantly (Results subsection “EEG-derived measure of confidence”, last paragraph; Figure 2E). Nevertheless, we also included a parametric regressor modulated by prestimulus alpha power in our fMRI GLM model in order to absorb potential variability associated with this signal.

Secondly, we focused on potential effects of sustained fluctuations in subjects’ attention (i.e., across trials). Specifically, we looked for correlations in the EEG-derived measures between neighbouring trials. We found that a serial autocorrelation analysis predicting component amplitudes Y based on the immediately preceding 5 trials provided limited explanatory power (subject-averaged R^2^ =.03) (Results, see the aforementioned paragraph). Overall, these observations suggest that our results are unlikely to be purely explained by attentional factors.

Other comments:In order to accord with requirements for reporting statistics the paper here should add a statement that "No explicit power analysis was conducted for determining sample size".

We have updated the text of the manuscript accordingly (subsection “Participants”).

Reviewer #4:[…] Overall this study offers new insights on the origins of confidence during perceptual decisions, by showing that the vmPFC also encodes an early confidence readout for this type of choices. I am a fan of the authors' methodological strategy to study human decision-making. However, for this study, my points of criticism are mainly related to the set of statistical analyses that the authors stand on to make their conclusions, which I consider should be revised. I provide some suggestions that may help to strengthen the authors' conclusions.1) An important concern is the statistical fMRI modelling approach. The delay between stimulus response and confidence rating is extremely short if one wants to incorporate the same parametric regressors at both the decision and confidence rating stage. I appreciate that there is a jitter of 1.5-4 s, however, given this short average duration (~2.6 s, roughly corresponding to a bit more than 1 TR), I suspect that adding the confidence rating as parametric modulator at both the rating and decision stage will be highly correlated. Even if FSL allows to run such model, highly correlated regressors (after convolution with the HRF) can have dramatic effects on the beta weights due to variance inflation (see for instance, Mumford et al., 2015).

We wish to clarify that the rating regressor at the time of decision (Ratings_DEC_) is locked to the onset of the random dot stimulus (i.e., rather than the behavioural response to the stimulus). Thus, the actual delay between the onsets of the decision-locked (Ratings_DEC_) and rating-locked (Ratings_RAT_) regressors is on average 3.84 (SD=.02) seconds. The jitter of 1.5-4 s that the reviewer is referring to spans only the time interval between the *end* of the response time window and onset of the rating prompt.

Relatedly, we note that in our experimental design, the timing of the inter-stimulus jitters was optimised using a genetic algorithm (Wager and Nichols, 2003) which served to increase estimation efficiency (we now report this in Materials and methods subsection “Main task”, second paragraph).

As per the reviewer’s suggestion we have now calculated the correlation between these two confidence rating regressors and show that they are only weakly correlated (mean R=-.13) (see Figure 5—figure supplement 3, top panel). To be fully transparent, we show the correlations for individual subjects and runs separately (Figure 5—figure supplement 3).

To more directly address this concern, we conducted two additional GLM analyses whereby only the regressors pertaining to one phase of the trial were included at a time (i.e. either the decision, or the rating, respectively). We found that activations for Ratings_DEC_, Ratings_RAT_, as well as Y_CONF_ remained qualitatively and quantitatively virtually identical to the original design that including both regressors (see Figure 5—figure supplement 4).

On a related issue, the authors write: "we also included a parametric regressor modulated by subjects' reaction time on the direction discrimination task (duration = 0.1 s, locked to the time of behavioural response)". First, did the authors also include the main effect regressor? This is not reported. If it wasn't included, then the model is wrongly specified. You cannot include a parametric regressor without including the main effect regressor. In any case, if this main effect regressor is indeed included, then once more, I suspect that this main effect regressor will be highly correlated with the main effect regressor that is included on trial onset. The mean response times are less than 1 s. If you convolve two stick functions that are less than 1 s apart, the resulting convolved regressors will be highly correlated.

Our model included a main effect regressor locked to the onset of the stimulus, which served to account for all parametric regressors in the decision phase of the trial, including the RT-modulated regressor (considering the short time span between RT and stimulus onset and the slow nature of the HRF).

We aimed to address the reviewer’s concern by running a separate GLM analysis which formally assessed whether including an unmodulated regressor at the time of RT would alter our results. We found that while this new regressor was indeed correlated with the unmodulated regressor at the time of stimulus onset (VSTIM_DEC_)(R=.73) as the reviewer speculates, activations for the Y_CONF_ regressor remained unchanged (see Author response image 1).

Positive parametric modulation of the BOLD signal by EEG-derived measures of confidence (during the decision phase of the trial), resulting from a GLM analysis whereby we included an additional unmodulated regressor locked to the time of response on the perceptual decision. Correlations with the EEG-derived confidence regressor have remained largely identical to those observed with the original GLM analysis (see Figure 4 for comparison). Results are reported at |Z|≥2.57, and cluster-corrected using a resampling procedure (minimum cluster size 162 voxels).

I would like to see (and I think this should be formally included as supplementary information in the manuscript) for the current design, an example of the design and design_cov figures produced by FLS for two or three subjects (or in general that the authors report the correlation not only between original the parametric regressors, but also the correlations between all the regressors including both main effects and parametric regressors after convolution with the HRF).

We thank the reviewer for this suggestion. We have now computed variance inflation factors for all regressors in our model and found that mean VIF = 3.57 ( ± 1.83), with multicollinearity typically being considered high if VIF > 5-10. We included these results in the manuscript (Materials and methods subsection “GLM analysis.”, last paragraph). We also illustrate the correlations between the identical confidence-related parametric regressors locked to the decision vs. rating stages of the trial (please see Figure 5—figure supplement 3), separately for each subject and experimental run (see response to earlier comment above).

Regarding the first point, I understand that the authors wanted to split explanatory variance between the decision and rating stages, but what I recommend and find more appropriate given my above-mentioned concern (if the authors insist in investigating the separation of confidence between the decision and the rating stages) is to run to separate GLMs, one with the parametric regressor on the decision stage, and one with the parametric regressor on the confidence rating stage and investigate whether the main conclusions of the authors still hold.

We have conducted the two suggested analyses using separate GLMs (please refer to our earlier comment). Our results remain quantitatively and qualitatively nearly identical (Figure 5—figure supplement 4).

Then, in a second level analysis the authors could investigate at which time point (decision or rating) there is a stronger relationship between the confidence ratings and BOLD responses.

We wish to clarify that our goal here was not necessarily to assess whether BOLD signals show stronger correlation with confidence ratings at one stage or the other. In fact, our results are particularly intriguing in that distinct neural networks appear to carry information about confidence during these two stages of the trial. In particular, activations during the decision phase of the trial such as the VMPFC or anterior cingulate cortex, appear consistent with a more automatic encoding of confidence, i.e., in the absence of explicit confidence report (Lebreton et al., 2015; Bang and Fleming, 2018). In line with this, we also observed activations in regions associated with the human reward/valuation system, such as the striatum and orbitofrontal cortex. In contrast, regions showing correlation with confidence during the confidence rating stage, in particular the anterior prefrontal cortex, have been previously associated with explicit metacognitive judgment/report (Fleming et al., 2012; Morales et al., 2018), perhaps serving a role in higher-order monitoring and confidence communication. We now address these points in the Discussion (third paragraph).

2) In the results presented in Figure 2B, if I understood correctly, the authors report the y(t) out-of-sample values of the "unseen" data for the middle confidence level. I am not convinced that the results reported by the authors are strong evidence for their decoder's ability to generate sensible out-of-sample y(t) values as this could simply reflect regression to the mean (if the authors use the middle confidence level for reading out y(t) by using a dichotomous decoder). I do not think that this result is especially revealing nor necessary to conduct the subsequent fMRI analysis (see my next point).

Please note that the results we report in relation to the Low- vs. High-confidence discrimination analysis (i.e., classifier performance) were based on a cross validation procedure to ensure there was no overfitting. Specifically, classifier performance (Az) for each subject was computed on the basis of Y values obtained from a leave-one-out procedure, whereby confidence-discriminating spatial filters (w) estimated using N-1 trials at a time were applied to the remaining trial(s) to obtain out-of-sample Y values. For clarity, we now describe this procedure in more detail in the revised paper (Materials and methods subsection “Single-trial EEG analysis”, third paragraph).

To address the reviewer’s concern, we compared these out-of-sample Ys with the values of Y obtained from the original Low- vs. High-confidence discrimination, and found that they were highly correlated (mean R value across subjects:.93, now reported in the aforementioned paragraph). Further, we repeated the analyses presented in Figure 2B, and found that on average, Y values for Medium-confidence trials continued to be situated between, and significantly different from, those in the Low-confidence (t(23)=-4.37, p<.001) and High-confidence (t(23)=-5.04, p<.001) trials; see new Figure 2—figure supplement. 2.

3) Regarding the use of the decoded value y(t) as parametric regressor for the fMRI analysis, if I understood correctly, the authors use out of sample values of y(t) only for the middle confidence level (see my concern in the point above), whereas for the low and high confidence levels this was not the case. I think that a more appropriate analysis would be to obtain values y(t) fully out of sample. The authors can split the data in n-folds (for instance 10) and use n-1 folds to train the data and obtain the y_CONF_ regressor using the remaining fold for decoding, and then use these y_CONF_ values as regressors of the fMRI data (see for instance for a similar n-fold cross-validation approach: van Bergen et al., 2015 Nature Neuroscience). Given the nature of the decoder used by the authors (dichotomous predictions), it should be enough that the authors split the data in high and low confidence levels to train the decoder (and therefore it is not necessary to use three levels or more).

We have now extracted out-of-sample values for the High- and Low-confidence levels resulting from a leave-one-trial-out procedure (please see previous point) and found that these values were highly correlated with the original Ys (mean R across subjects =.93), which we take as evidence for the generalisability of our decoder. Repeating the main GLM analysis using these values yielded nearly identical results (Figure 5—figure supplement 2).

Note that the use of 3 confidence bins (i.e., performing the EEG classification analysis using the extreme ends of the confidence rating scale) was done in an effort to increase sensitivity of the classification analysis and obtain more reliable discrimination weights (that is, minimise overlap between internal representations of High vs. Low confidence caused by potential within-subject inconsistency in confidence ratings), which would in turn improve the quality of our EEG-informed fMRI results. For this reason, we opted to perform the above analysis using the original trial split. Finally, please note that due to the limited number of trials per subject in the current experiment (≤320), we avoided estimating neural signals on subsets of the data in order to preserve the reliability of our results.

4) Subsection “EEG-derived measure of confidence”, last paragraph: Maybe I missed the point, but for me it is not entirely clear why it is expected that the discriminant component amplitudes are not different for correct and incorrect answers. One of the well established statistical signatures of confidence is that confidence is markedly different for correct and incorrect responses (e.g. see Sanders, Hangya and Kepec, 2016; Urai, Braun and Donner, 2017). Why didn't the authors expect a separation (see my next point for a related concern), and if not, what the discriminant component amplitude really reflects? More discussion on this point in general would be great.

This is an important consideration and we thank the reviewer for pointing it out. The goal here was to ensure that variability in discriminant component amplitudes were not driven solely by fluctuations in decision accuracy. For example, one could argue that the confidence discrimination patterns we observe in the EEG might be purely explained by an unbalanced proportion of correct and error responses in the confidence trial bins used for discrimination (e.g., more correct trials in the High-confidence bin than the Low-confidence bin). We wanted to demonstrate that even when accuracy is constant (i.e., when correct and error trials are considered separately) we continue to see significant effects of confidence in the discriminant amplitude Y (Figure 2D). However we agree that the lack of an effect of accuracy on Y calls for additional investigation. We ran a correlation analysis to assess this relationship and found a small but significant positive correlation between accuracy and Y (t-test on regression coefficients: t(23)=8, p<.001). The original analysis might have been less sensitive to this effect due to the binning of trials by both confidence and accuracy, which meant that mean Y estimates per bin were made from only a few trials (≤10) in some cases. We report this new analysis in the revised paper (Results subsection “EEG-derived measure of confidence”, sixth paragraph) and have also added a new panel which illustrates this relationship (Figure 2C).

More broadly, the point we wish to make is that while our EEG-derived neural measures of confidence might correlate partly with performance (i.e., choice accuracy) as would be expected from existing work and as pointed out by the reviewer, it can more importantly be decoupled from it. Indeed, much of recent work supports the idea of a dissociation between performance and confidence/metacognition (Lau and Passingham, 2006; Rounis et al., 2010; Komura et al., 2013; Lak et al., 2014; Fleming and Daw, 2017).

5) Subsection “Stimuli and task”, last paragraph: On a related point (and perhaps an important caveat of this study), I do not understand why the authors excluded or explicitly asked the participants to "abstain from making a confidence response on a given trial if they became aware of having made an incorrect response". Again, one of the well established signatures of confidence is that confidence is markedly different for correct and incorrect responses. How the results would have been affected without such explicit instruction to the participants? In my opinion, this confidence information should not be excluded or spuriously biased via instructions to the participants. Therefore, I am afraid that what the authors are capturing with their actual confidence ratings (and therefore the decoded values y(t)) is a biased response that is not formally confidence per se. I urge the authors to report this potential caveat and make a clear case (from the beginning) of why this strategy was adopted in the first place.

We apologise for the ambiguity in the description of the task instructions. To clarify, subjects were instructed to refrain from making a confidence rating only if a *motor mapping* error had been made, for example a premature response that was accidentally initiated in favour of one motion direction despite clear perceptual representation of the opposite direction (this was reported by some subjects following practice sessions, and in previous experiments). Our motivation for wanting to exclude these trials from our analyses was that we were interested in the representations of confidence associated with the perceptual judgment/choice per se, as opposed to the physical action that accompanied it.

As there was no strong emphasis on speed in our paradigm (subjects were allowed up to 1.3s to make a response) we anticipated that this would be a relatively rare occurrence. Indeed, the number of trials in which no confidence rating was recorded following a perceptual choice was very small, on average 6.13 (SD=5.4) trials per subject, representing only 1.9% (SD=1.7%) of the total number of trials, thus suggesting that this instruction could not have had a substantial impact on our results.

We have amended the revised paper to clarify the description of the task instructions (see Materials and methods subsection “Main task”).

[Editors' note: the author responses to the re-review follow.]

Major comments:1) One issue is that the model makes the same predictions for every trial, as task difficulty (motion coherence) is held constant. So the authors are bound to between-subject correlations, which they found indeed between modeled sensory evidence at the bound (Δe) and EEG-derived predictor of confidence rating (Y). My understanding of the current interpretation is that Y represents evidence plus some neural noise, and then that confidence rating (R) is Y plus something else (perhaps noise again) which could be loosely defined as 'meta-cognitive reappraisal'. This is not very informative, and not even directly tested.A more straightforward test would be a mediation analysis, assessing whether Y could indeed mediate the link from Δe to R. The alternative hypothesis to be discarded is that Δe is actually closer to R, which would take us back to the fundamental question of how Y can be specified in cognitive terms.Yet a more informative use of the race model would be to fit trial-by-trial variations in R. This means allowing free parameters to vary across trials, and to test their potential relationship with Y. It could be for instance that Y fluctuations arise from variations in the starting point, or, within a Bayesian framework, in the prior on motion direction. Having said this, my agenda is not to bury the paper under requests for additional work. A clarified relationship between Δe, Y and R may be a reasonable limit to what can be inferred from the dataset.

We thank the reviewer for their constructive feedback. We have now carried out the suggested mediation analysis and report these results in the paper (subsection “Exploratory mediation analysis”). We have also made changes to the Discussion where we attempt to clarify the relationship between Δe, Y, and Ratings, whilst acknowledging current limitations in interpretation and highlighting some remaining open questions (Discussion, fifth to eleventh paragraphs).

Please note that as our current computational model does not provide trial-to-trial correspondence between model data (Δe) and observed data (Y/Ratings), the mediation analysis was performed at the subject group level (Materials and methods subsection “Exploratory mediation analysis”). As in our previous analysis linking Y and Δe (Figure 3D), we first computed the mean difference between correct and error trials for each of the three variables of interest, to produce measures that are comparable across subjects (i.e., by removing individual differences in the trial-averaged scores that may be due to task-irrelevant factors, e.g., rating biases). These quantities (henceforth referred to as Δe_DIFF_, Y_DIFF_, and Ratings_DIFF_) were then subjected to a mediation analysis testing the hypothesis that Y mediates the link between Δe and ratings. Specifically, we defined a three-variable path model (Wager et al., 2008) with delta-E_DIFF_ as the predictor variable, Ratings_DIFF_ as the dependent variable, and Y_DIFF_ as the mediator. Consistent with the initial prediction, we found that: 1) delta-E_DIFF_ was a significant predictor of Y_DIFF_ (p=.01), 2) Y_DIFF_ reliably predicted Ratings_DIFF_ after accounting for the effect of predictor deltaE_DIFF_ (p<.001), and 3) the indirect effect of Y_DIFF_, defined as the coefficient product of effects 1) and 2), was also significant (p=.004) (Results subsection “Exploratory mediation analysis”).

Indeed, the mediator effect of the EEG-derived confidence is in agreement with the idea that Y represents a (potentially noisy) readout of decision-related balance of evidence (as modelled by Δe) by the vmPFC, which in turn serves as a basis for subjective confidence ratings. Please note, however, that given the across-subject nature of the analysis, these findings must be interpreted with some caution.

In conjunction with the observations presented in the manuscript, we interpret Y as largely relying on, though potentially distinct from, the quantity represented by Δe, in line with the idea of a dissociation between the information that supports the decision vs. confidence. Importantly, the timing of Y (i.e., which is recorded prior to commitment to a motor response), suggests that these early neural estimates of confidence arise in close temporal proximity to the decision, in line with an automatic readout of confidence (i.e., in the absence of explicit report) (Lebreton et al., 2015) and the proposal that vmPFC might encode an early and automatic “feeling of rightness” (Moscovitch and Winocur, 2002; Hebscher and Gilboa, 2016) in memory judgments. While dedicated research will be necessary to establish the functional role of this quantity, early/fast pre-response confidence signals could be necessary to regulate the link between decision and impending action, e.g. with low confidence signalling the need for additional evidence (Desender et al., 2018). We now make these points more explicit in the Discussion (eighth paragraph).

Regarding the link between Y and the neural activity leading to explicit confidence ratings, it has been long proposed that metacognitive evaluation relies on additional post-decisional processing (Pleskac and Busemeyer, 2010; Moran et al., 2015; Yu et al., 2015). For instance, recent evidence suggests that choice itself (and corresponding motor-related activity) impacts confidence (Fleming et al., 2015; Gajdos et al., 2018) and may help calibrate/optimise metacognitive reports (Siedlecka et al., 2016; Fleming and Daw, 2017). In this framework, Y could serve as one of multiple inputs to networks supporting retrospective metacognitive processes such as the anterior prefrontal regions (Fleming et al., 2012), which would explain both the correlation with, and dissociation from, subjects’ confidence reports (Discussion, tenth paragraph).

2) A related issue comes with the new analysis provided to substantiate a role for confidence in behavioral control. The authors found that higher confidence predicts repetition of the same choice in the next trial, if motion direction is the same as in the current trial. This is not in line with the computational model in its present form. It could mean that confidence influences the prior on motion direction, but this would obviously not be adaptive. There is therefore a need to reconcile this finding with the generative model of choice and confidence.

Our analyses investigating the influence of neural confidence signals (Y) on subsequent behaviour indicated that high Y amplitude increased the likelihood of repeating a choice only if stimulus direction was consistent with that of the previous trial. Crucially, we did not see this repetition bias when the subsequent trial showed a stimulus moving in the opposite direction. This dependence of choice repetition on stimulus identity is not straightforward to interpret/model (indeed, we are not aware of similar observations in the literature), as it suggests the existence of a potentially separate process which detects consistency between the previous and current stimulus, and which interacts with the representation of previous confidence to influence decision/behaviour (e.g., through selective re-weighting of evidence).

As such, our results cannot be explained by a direct effect of confidence on the motion prior/starting point, as this would predict a generic repetition bias impacting subsequent choices equally, regardless of whether the next presented stimulus motion is in the same or opposite direction. Indeed, we confirmed this hypothesis with a new analysis using a modified version of the model whereby starting point on a given trial was biased in the direction of the previous choice, in proportion to the magnitude of Δe associated with that choice.

If indeed stimulus congruency interacts with the representation of previous confidence to influence decision/behaviour via a separate process one might not necessarily be able to capture the choice repetition bias as observed here only by modelling the decision process itself. Therefore, for the purposes of this paper, we designed our model to describe the decision process without accounting for potential trial-to-trial dependencies arising from the interaction with confidence, to offer an initial mechanistic interpretation of Y and provide a plausible link to the eventual confidence ratings. We now acknowledge this point in the Discussion (sixth paragraph) and highlight the need to reconcile our findings with formal models of decision making.

Another potential interpretation of this finding is that it is used as a means for balancing speed/accuracy demands in rapid decision tasks, even if this might inevitably lead to sub-optimal (local) behaviour. Since our task was not specifically designed to manipulate such variables (e.g. transitions in stimulus identity, speed/accuracy etc.) we cannot, at this stage, offer unequivocal support to this idea – though we have started to think carefully of possible extensions of this work and we are beginning to plan future experiments accordingly.

We hope our results will serve as a starting point for future work that will be tailored specifically towards sequential trial dependencies between neural confidence and stimulus/decision dynamics.

3) Y is not different for correct/incorrect responses: The authors should add a figure plotting Y split for correct and incorrect responses (perhaps next to the new panel 2C) indicating the quantitative difference. Also, please add an explicit explanation (in the Results/Discussion section) about this result. This is essential, as I believe it reveals quite a lot about what type of information Y carries, namely, it is not the classical statistical signature of confidence (see Sanders and Urai work) but something different in line to the arguments that the authors give at the end of the response to this point, i.e. not the probability that the choice is correct, but something else. The authors try to describe this "dissociation" to some extent in other parts of the text, but should be more explicit about this point.

We have now added a new panel (Figure 2D) showing the trial-averaged Y for correct and incorrect responses. We quantified this difference and found that across subjects, Y was consistently higher for correct than error responses (t(23)=7.58, p<.001, Results subsection “EEG-derived measure of confidence”, sixth paragraph), in line with the assumption that these early confidence signals are at least partially informed by the same information leading to the decision. That said, the observation that effects of confidence in the discriminant amplitude Y remain significant when accuracy is kept constant (i.e., when correct and error trials are considered separately) could indicate that, as the reviewer suggests, a dissociation of the confidence-related signals from statistical confidence exists. Y may be more in line with what has been referred to as “subjective confidence” (Odegaard et al., 2018), i.e., a quantity that can largely track statistical confidence (Sanders et al., 2016) but be prone to additional discrepancy or bias. Additional work could help directly address this question, for example by explicitly manipulating perceived confidence independently of task performance (Odegaard et al., 2018). The nature of discrepancies between statistical and subjective confidence has been addressed in behavioural and modelling studies (e.g., (Zylberberg et al., 2012; Samaha et al., 2018), however it is less well understood at the neural level, and lies beyond the main scope of this paper. We have adapted the Discussion to make the above points mode explicit (sixth paragraph).

4) Please add the results of the neural correlates of Y at the rating stage (can be in the supplement), and briefly comment about this in the Results section. Reviewers understood that the authors did not want that to focus on this stage, but it is quite a nice result that information about Y transitions from vmPFC at the decision stage to motor related areas at the confidence rating stage.

We have now included these results in the manuscript, along with the brief interpretation as suggested by the reviewer (Results subsection “fMRI correlates of EEG-derived confidence signals”, last paragraph).

References:

Dutilh G, Vandekerckhove J, Forstmann BU, Keuleers E, Brysbaert M, Wagenmakers EJ (2012) Testing theories of post-error slowing. Attention, perception & psychophysics 74:454-465.

Moscovitch M, Winocur G (2002) The frontal cortex and working with memory. Principles of frontal lobe function:188-209.

Odegaard B, Grimaldi P, Cho SH, Peters MAK, Lau H, Basso MA (2018) Superior colliculus neuronal ensemble activity signals optimal rather than subjective confidence. Proceedings of the National Academy of Sciences of the United States

of America 115:E1588-E1597.

Samaha J, Switzky M, Postle BR (2018) Confidence boosts serial dependence in orientation estimation. bioRxiv.

Sanders JI, Hangya B, Kepecs A (2016) Signatures of a Statistical Computation in the Human Sense of Confidence. Neuron 90:499-506.

Zylberberg A, Barttfeld P, Sigman M (2012) The construction of confidence in a perceptual decision. Frontiers in integrative neuroscience 6:79.